# The coordinated action of the MVB pathway and autophagy ensures cell survival during starvation

Martin Müller[1†], Oliver Schmidt[1†], Mihaela Angelova[2], Klaus Faserl[3], Sabine Weys[1], Leopold Kremser[3], Thaddäus Pfaffenwimmer[4], Thomas Dalik[5], Claudine Kraft[4], Zlatko Trajanoski[2], Herbert Lindner[3], David Teis[1,6*]

[1]Division of Cell Biology, Biocenter, Medical University of Innsbruck, Innsbruck, Austria; [2]Division of Bioinformatics, Biocenter, Medical University of Innsbruck, Innsbruck, Austria; [3]Division of Clinical Biochemistry, ProteinMicroAnalysis Facility, Biocenter, Medical University of Innsbruck, Innsbruck, Austria; [4]Max F. Perutz Laboratories, University of Vienna, Vienna, Austria; [5]Department of Chemistry, University of Natural Resources and Applied Biosciences, Vienna, Austria; [6]Austrian Drug Screening Institute, Innsbruck, Austria

**Abstract** The degradation and recycling of cellular components is essential for cell growth and survival. Here we show how selective and non-selective lysosomal protein degradation pathways cooperate to ensure cell survival upon nutrient limitation. A quantitative analysis of starvation-induced proteome remodeling in yeast reveals comprehensive changes already in the first three hours. In this period, many different integral plasma membrane proteins undergo endocytosis and degradation in vacuoles via the multivesicular body (MVB) pathway. Their degradation becomes essential to maintain critical amino acids levels that uphold protein synthesis early during starvation. This promotes cellular adaptation, including the de novo synthesis of vacuolar hydrolases to boost the vacuolar catabolic activity. This order of events primes vacuoles for the efficient degradation of bulk cytoplasm via autophagy. Hence, a catabolic cascade including the coordinated action of the MVB pathway and autophagy is essential to enter quiescence to survive extended periods of nutrient limitation.

*For correspondence: david. teis@i-med.ac.at

†These authors contributed equally to this work

Competing interests: The authors declare that no competing interests exist.

## Introduction

Evolutionary conserved selective and non-selective protein degradation pathways are essential for cell growth and survival. The ubiquitin-proteasome system (UPS) mediates selective poly-ubiquitination of cytoplasmic proteins and their degradation at 26S proteasomes for regulatory and quality control functions. Mis-folded proteins in the endoplasmic reticulum (ER) are also ubiquitinated, extracted by the ER-associated protein degradation system (ERAD) and degraded at 26S proteasomes in the cytoplasm (*Vembar and Brodsky, 2008*).

Macro-autophagy (hereafter autophagy) non-selectively transports bulk cytoplasm into lysosomes. Therefore the induction of autophagy is tightly controlled: under normal growth conditions autophagy operates on a basal level because it is suppressed by signaling from the target of rapamycin complex 1 (TORC1) (*Kamada et al., 2000*; *Loewith and Hall, 2011*; *Zoncu et al., 2011b*). In response to cellular stress, such as nutrient depletion (e.g., of amino acids), TORC1 is inactivated (*Loewith and Hall, 2011*) and autophagy is strongly induced. Deregulation of autophagic processes is implicated in metabolic and infectious diseases as well as in cancer or neurodegeneration (*Rubinsztein et al., 2012*). Once induced, the autophagic machinery begins to sequester cytoplasmic components,

**eLife digest** Yeast and other organisms have evolved to survive extended periods of starvation by digesting their own proteins and other cell materials and thereby recycle them into new proteins and structures. One way in which these cell materials can be destroyed is by a process called autophagy. A membrane forms around the cell material to isolate it from the rest of the cell. In yeast, the resulting structure fuses with a cell compartment called the vacuole, which contains enzymes that break down the cargo into smaller molecules that can be re-used by the cell.

When cells experience starvation, autophagy is not very selective in what it destroys and so it is tightly controlled to avoid damaging important structures in healthy cells. Alongside autophagy, specific proteins in the membrane surrounding a yeast cell can be targeted for destruction by another process called the MVB pathway. Certain membrane proteins are tagged with a small protein called ubiquitin, which leads them to being selectively incorporated into cell compartments called MVBs that then go on to fuse with the vacuole. However, it is not clear how the MVB pathway and autophagy may cooperate to enable the cell to survive periods of starvation.

Here, Müller et al. monitored the changes in the proteins present in yeast cells during a period of starvation. The experiments show that many different membrane proteins in the yeast cells were destroyed via the MVB pathway within three hours of the removal of their food source. This was essential to allow the cells to carry on producing new proteins at this early stage in starvation. These new proteins included the enzymes found in vacuoles, which increased the ability of the cells to break down the proteins and other cell materials that were transported there via autophagy.

These findings show how the MVB pathway and autophagy are co-ordinated to allow cells to survive periods of starvation. The next challenge is to work out how the MVB pathway is regulated at the molecular level in response to fluctuations in nutrient availability.

ribosomes and organelles within a large double-membrane compartment termed the autophagosome (*Yang and Klionsky, 2010*; *Kraft and Martens, 2012*; *Mizushima et al., 2011*). In addition, some core components of the autophagic machinery such as LC3/Atg8 are transcriptionally induced (*Kirisako et al., 1999*). Direct fusion of autophagosomes with lysosomes delivers autophagic bodies and the sequestered cargo into the lysosomal lumen. Alternatively, autophagosomes can first fuse with multivesicular bodies (MVBs) to form so-called amphisomes, before they fuse with lysosomes (*Seglen et al., 1991*). Finally, the breakdown of autophagic bodies and the efficient degradation of autophagic cargo inside lysosomes is required to recycle amino acids, nucleotides, carbohydrates and lipids back to the cytoplasm. The recycling of these key metabolic building blocks protects cells from their fatal depletion and thus maintains cellular homeostasis to survive nutrient limitation (*Onodera and Ohsumi, 2005*; *Vabulas and Hartl, 2005*; *Jones et al., 2012*; *Suraweera et al., 2012*). Therefore evolutionary conserved starvation programs in mammalian cells and yeast expand and strengthen this intracellular recycling system by enhancing the de novo synthesis of vacuolar/lysosomal hydrolases (*Gasch et al., 2000*; *Sardiello et al., 2009*; *Settembre et al., 2011*; *Shen and Mizushima, 2014*).

In addition to autophagy, TORC1 also regulates ubiquitin-mediated endocytosis of integral plasma membrane proteins. On the one hand, TORC1 signaling was required to promote the endocytosis of certain plasma membrane proteins (*MacGurn et al., 2011*). On the other hand, inactivation of TORC1 either by rapamycin or starvation, triggered the endocytosis of other plasma membrane proteins that were subsequently degraded in an ESCRT (endosomal sorting complex required for transport)-dependent manner via the MVB pathway (*Schmidt et al., 1998*; *Jones et al., 2012*; *Lang et al., 2014*). The extent to which starvation induces plasma membrane remodeling has yet to be determined. Furthermore, how the subsequent ubiquitin-dependent degradation of membrane proteins via the MVB pathway helps to meet the specific metabolic and energetic demands of cells during nutrient limitation is not fully understood. Therefore it is also not clear how selective (MVB) and non-selective (autophagy) lysosomal proteolysis pathways cooperate to mediate cell survival during nutrient limitation.

To comprehensively address these questions we have used quantitative proteomics. Our results demonstrate that within the first 3 hr of amino acid starvation many integral plasma membrane proteins, including high-affinity amino acid permeases, glucose transporters and G-protein coupled

receptors, were selectively removed from the cell surface by endocytosis and subsequently targeted into vacuoles via the ESCRT-dependent MVB pathway and degraded, while others remained stable or were up-regulated (e.g., the general amino acid permease, Gap1). This comprehensive and selective remodeling of the plasma membrane appeared to be completed within 3–4 hr of starvation. Autophagy was also immediately activated upon starvation and remained active throughout starvation. Surprisingly, early during starvation the selective degradation of membrane proteins via the MVB pathway was mainly responsible to maintain critical levels of free intracellular amino acids that were sufficient to uphold protein synthesis and promote the corresponding adaptation of the proteome. Most notably this included the de novo synthesis of vacuolar hydrolases, which boosted the proteolytic activity of vacuoles to support the efficient degradation of autophagic cargo. The continuous delivery and degradation of autophagic cargo further enhanced intracellular amino acid recycling and was ultimately essential to restore intracellular amino acid pools of cells during extended starvation. These findings reveal an unexpected role for the MVB pathway in maintaining intracellular amino acid homeostasis and thereby promoting the up-regulation of vacuolar hydrolases early during starvation, which is tightly coordinated with autophagy. This catabolic cascade is ultimately required to allow starving cells to complete their cell division cycle and enter a quiescent state for survival.

## Results

### Starvation induces selective and non-selective protein degradation pathways

To understand how the MVB pathway, autophagy and proteasomal degradation cooperate during nutrient limitation, we first analyzed the starvation-induced degradation of model proteins in yeast.

To assess selective membrane protein degradation via the MVB pathway, we followed the ubiquitin-dependent endocytosis of the plasma membrane methionine permease, Mup1-GFP and its transport into the vacuole in response to starvation (for amino acids and nitrogen sources) (*Beck et al., 1999*; *Menant et al., 2006*; *Jones et al., 2012*). Under rich growth conditions Mup1-GFP is mainly found at the plasma membrane and very little is degraded (*Figure 1A,B*). Yet, within 3 hr after the onset of starvation the majority of Mup1-GFP was removed from the cell surface, delivered into vacuoles and degraded (*Figure 1A,B*). The proteolytic degradation of Mup1-GFP inside vacuoles released free GFP, which remained stable and was monitored by western blotting (*Figure 1A*). The starvation-induced delivery of Mup1-GFP into the vacuole was dependent on the ESCRT machinery but was not affected in an autophagy (*atg8Δ*) mutant (*Figure 1B*). In an ESCRT (*vps4Δ*) mutant, the MVB pathway was blocked and Mup1-GFP was not delivered into the vacuole but instead accumulated on the class E compartment and at the plasma membrane (*Figure 1B*).

To define the timing of starvation-induced degradation of Mup1-GFP in the context of eukaryotic starvation programs, we compared it to the delivery of bulk cytoplasm via autophagy. Therefore we determined the degradation of highly abundant selective (ribosomes) and non-selective (Fba1) autophagic cargoes. Growing yeast cells contain about 200,000 ribosomes that occupy up to 30–40% of the cytoplasmic volume (*Warner, 1999*). Upon starvation, otherwise stable ribosomes are among the first autophagic cargoes and rapidly degraded by selective (ribophagy) and non-selective autophagy (*Kraft et al., 2008*; *Ossareh-Nazari et al., 2014*). We monitored the release of free GFP from two different ribosomal proteins by western blotting: Rpl25-GFP (large subunit) and Rps2-GFP (small subunit). Both are fully functional GFP fusion proteins that incorporate into ribosomes (*Kraft et al., 2008*). When equal amounts of cell lysates were subjected to western-blot analysis, the protein levels of full length Mup1-GFP and the GFP-tagged ribosomal subunits were comparable (*Figure 1A*, lanes 6, 16). After 3 hr, at a time when the majority of full length Mup1-GFP was already degraded, free GFP from Rpl25 was first detected, showing that autophagy was also delivering cytoplasmic contents into the vacuole (*Figure 1A*, lane 8). During subsequent starvation the protein levels of free GFP from both ribosomal subunits increased. Monitoring the autophagy-dependent degradation of Fba1-GFP, one of the most abundant cytoplasmic proteins with approximately 1.000.000 molecules/cell (*Ghaemmaghami et al., 2003*), yielded similar results. Free GFP was first detected after 3 hr of starvation and the protein levels free GFP strongly increased during subsequent starvation (*Figure 1—figure supplement 1A*). To determine the earliest possible starvation-induced autophagic activity, we monitored the transport and degradation of fully functional GFP-Atg8. Atg8 is a core

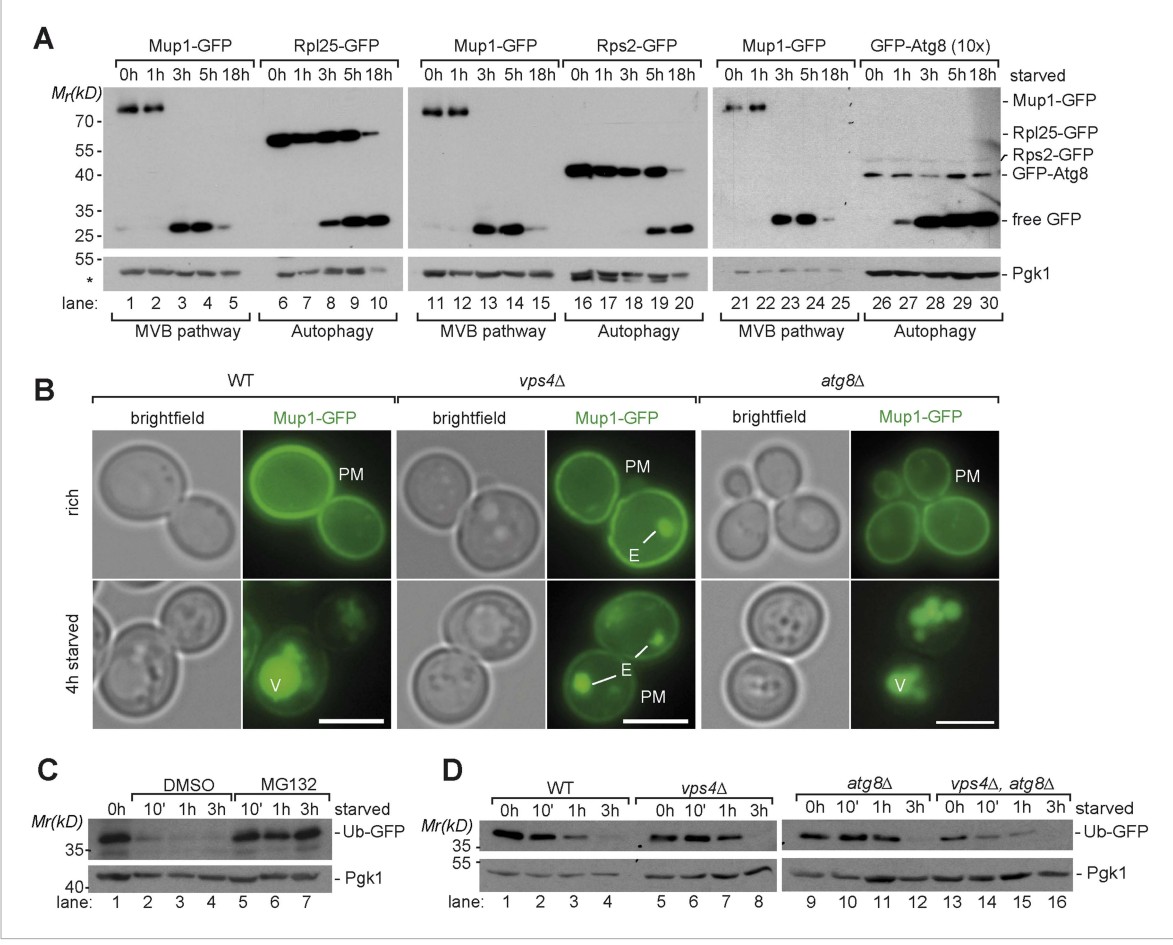

**Figure 1**. Starvation induces selective and non-selective protein degradation pathways. (**A**) WT cells expressing Mup1-GFP, Rpl25-GFP, Rps2-GFP or GFP-Atg8 were grown in rich medium (0 hr) or starved as indicated. Cell lysates were analyzed by SDS-PAGE and western blot (WB) using the indicated antibodies. *residual anti-GFP signal after re-probing the membrane with anti-Pgk1 antibody. (**B**) Fluorescence microscopy of Mup1-GFP in WT cells, *vps4Δ* mutants and *atg8Δ* mutants growing under rich or starvation conditions. (V)acuoles, (P)lasma (M)embrane and class (E) compartments. Scale bar = 5 μm. (**C**, **D**) Whole cell lysates of WT cells or the indicated mutants grown under rich conditions or starved for the indicated times were separated by SDS-PAGE and analyzed by western blot using the indicated antibodies. (**C**) *pdr5Δ* cells were treated with the proteasome inhibitor MG132 (50 μM) or vehicle (DMSO) during starvation.

The following figure supplement is available for figure 1:

**Figure supplement 1**. Induction of autophagy.

component of the autophagic machinery that remains conjugated to the inner membrane of all selective and non-selective autophagosomes, including <u>c</u>ytoplasm to <u>v</u>acuole <u>t</u>argeting (cvt)-vesicles. Therefore Atg8 is degraded together with autophagic cargo inside vacuoles. To be able to compare the degradation of GFP-Atg8 to Mup1-GFP, 10 times more lysate of cells expressing GFP-Atg8 was subjected to western blot analysis (*Figure 1A*). Small amounts of free GFP released from GFP-Atg8 inside vacuoles could be readily detected by western blot analysis 1 hr after the onset of starvation and the levels of free GFP strongly increased at 3 hr of starvation (*Figure 1A*, lane 27–30). These findings are consistent with the strong increase of endogenous Atg8 levels during starvation (*Figure 1—figure supplement 1B*) as observed earlier (*Kirisako et al., 1999*). Previous work also demonstrated that Atg8 protein levels control the size of autophagosomes but not the frequency (about 9 autophagosomes/hour) by which they are formed (*Abeliovich et al., 2000*; *Xie et al., 2008*). Hence, the increase in Atg8 protein levels during the first 4 hr of starvation would result in the formation of bigger (but not more) autophagosomes that could capture larger volumes of cytoplasm later during starvation. Our results for the early degradation of GFP-Atg8 and the continuous increase

in autophagic degradation of highly abundant selective as well as non-selective cargoes throughout starvation are fully consistent with this model. This idea was further supported using the Pho8Δ60 assay, a sensitive method to measure bulk autophagy (*Noda et al., 1995*). Pho8Δ60 activity was low under rich conditions, began to increase during the first 3 hr of starvation and continuously increased during extended periods of starvation (*Figure 1—figure supplement 1C*). These results show that autophagy is immediately activated upon starvation and delivers increasing volumes of cytoplasmic material into the vacuole with ongoing starvation (*Figure 1*).

Additionally, we investigated how cytoplasmic proteins were degraded at proteasomes upon starvation. Therefore, we employed a ubiquitin-GFP (Ub-GFP) fusion protein, which is an established reporter for proteasomal activity (*Johnson et al., 1992*; *Vabulas and Hartl, 2005*). It is detected at low levels in proliferating cells reflecting the equilibrium between its rapid degradation and its synthesis. Upon starvation, Ub-GFP was rapidly degraded at 26S proteasomes. The degradation of the reporter was exclusively dependent on proteasomal degradation but did not require autophagy or the MVB pathway (*Figure 1C*, lanes 5–7; *Figure 1D*).

Overall, these findings indicate that starvation triggered protein degradation by different selective and non-selective degradation pathways: the constitutive protein degradation via the proteasome was active from the onset of starvation and was previously suggested to play a key role upon acute nutrient restriction (*Vabulas and Hartl, 2005*). Our results further suggest that both autophagy and starvation induced-endocytosis were simultaneously activated early during starvation. Autophagy continuously delivered ever-increasing volumes of cytoplasm into vacuoles, whereas the starvation-induced degradation of membrane proteins was completed within 3 hr. These findings suggest an important role for the MVB pathway early during starvation.

## The MVB pathway and autophagy contribute differentially to maintain free amino acid levels and protein synthesis during starvation

While protein degradation at 26S proteasomes provides an immediate amino acid pool for protein synthesis already within minutes of acute starvation (*Vabulas and Hartl, 2005*) and autophagy is required to supply amino acids during extended periods of starvation (*Onodera and Ohsumi, 2005*), the relative contribution of the MVB pathway to overall amino acid homeostasis was not clear.

Therefore we next measured the intracellular levels of 18 different amino acids in isogeneic WT cells, MVB (*vps4Δ*) or autophagy (*atg8Δ*) mutants as well as double mutants (*vps4Δ, atg8Δ*) by liquid chromatography (*Altmann, 1992*). These strains were auxotrophic for the amino acids lysine and leucine. When grown in synthetic medium supplemented with amino acids (rich), the intracellular free amino acid levels were comparable in WT cells and autophagy mutants (*atg8Δ*) (*Figure 2A*), but slightly lower in *vps4Δ* mutants, which was mainly due to reduced lysine and arginine levels (*Figure 2A,B*).

1 hr after starvation in synthetic medium without amino acids and ammonium salts, the total free amino acid pool decreased to similar levels in WT cells and all mutant strains (*Figure 2A,B*). In WT cells the levels of most amino acids continued to decrease for another hour. Interestingly, at around 4 hr of starvation the overall levels of amino acids almost fully recovered, suggesting strong amino acid recycling. However, the levels of arginine and lysine, which were among the most abundant free amino acids, decreased further. The levels of glutamine, threonine and glycine did not recover very well, while the levels of other amino acids (particularly of glutamate) increased. The recovery of amino acid levels after 4 hr of starvation was strongly dependent on autophagy. These results are at large consistent with previous findings, where an approximately threefold reduction in intracellular amino acid levels was detected during the first 2 hr of starvation and autophagy was required for the partial recovery of amino acid levels from 3 to 6 hr of starvation (*Onodera and Ohsumi, 2005*). In our strain background the levels of amino acids were generally lower under rich growth and we observed an approximately twofold reduction in intracellular amino acids during the first 2 hr of starvation. At 4 hr of starvation amino acid levels fully recovered in an autophagy dependent manner (*Figure 2A,B*).

In addition, our findings showed that the MVB pathway essentially contributed to maintain the overall levels of free intracellular amino acids. 1 hr after starvation, the amino acids levels decreased similar in *vps4Δ* mutants, autophagy mutants (*atg8Δ*) and WT cells. However, after 2 hr the overall amino acid levels were lower in *vps4Δ* mutants compared to WT cells and autophagy mutants. The levels of 14 individual amino acids were lower in *vps4Δ* mutants when compared to WT cells or autophagy mutants. Moreover the amino acid levels failed to recover during extended starvation

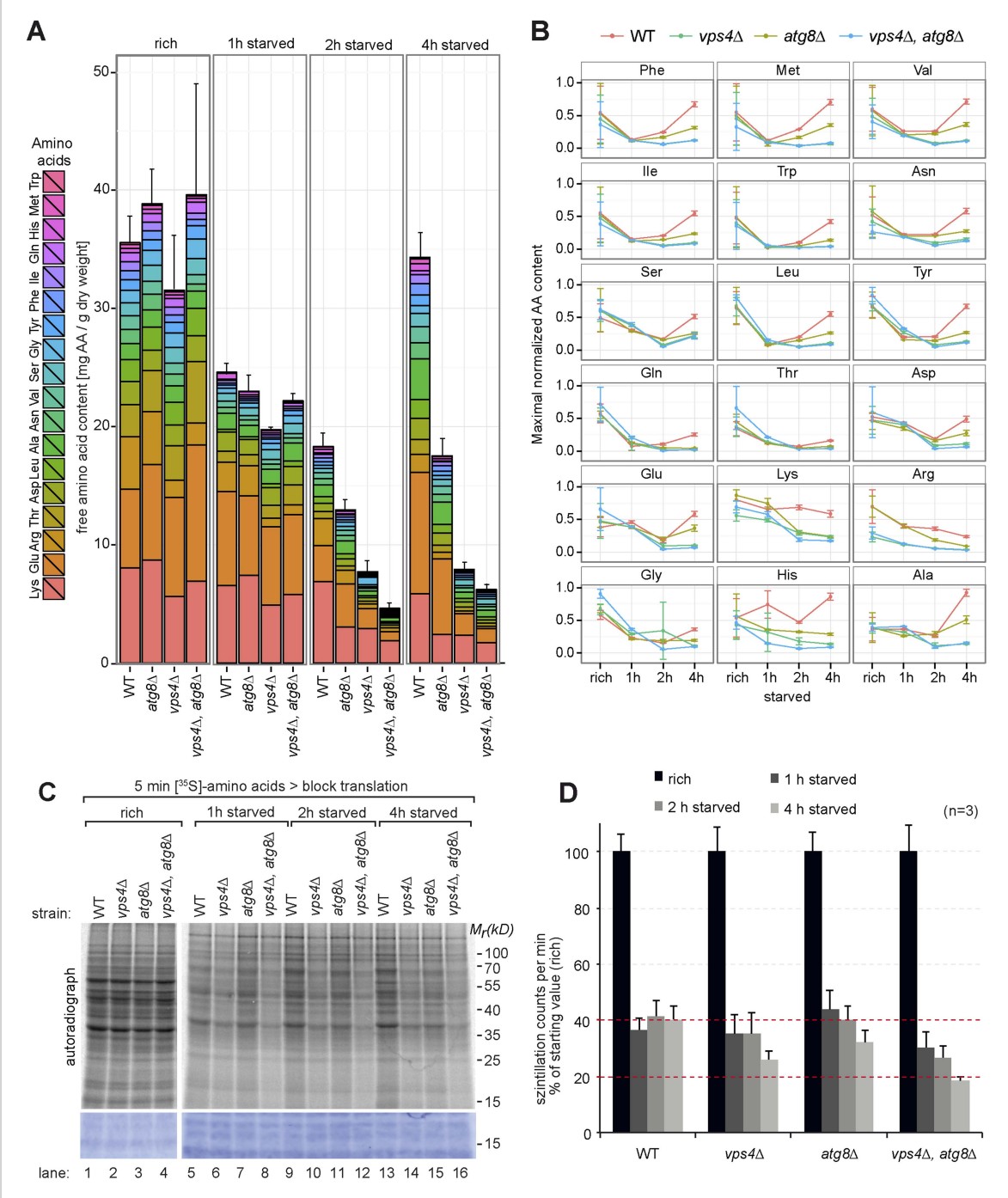

**Figure 2**. Changes in free amino acid levels and protein synthesis during starvation. (**A**) Cells were grown to mid-log phase (rich) and starved as indicated. Free amino acids were extracted and analyzed by liquid chromatography. Data are represented as the sum of free amino acids (mg) per gram of dry yeast. Mean ± SD, n ≥ 3. (**B**) Changes in individual amino acids from (**A**) normalized to maximal values. Mean ± SD, n ≥ 3. (**C, D**) Cells grown under the indicated conditions were incubated for 5 min with $^{35}$S-labeled Met and Cys. (**C**) $^{35}$S-incorporation was analyzed by SDS-PAGE and digital autoradiography. Coomassie staining shows equal protein loading. (**D**) Quantification of $^{35}$S-incorporation under rich conditions and after 1, 2 and 4 hr of starvation by liquid scintillation counting. Incorporation under rich conditions was set to 100%. Mean ± SEM, n = 3.

The following figure supplement is available for figure 2:

**Figure supplement 1**. Changes in free amino acids levels during starvation in prototrophic yeast.

(*Figure 2A*). From 2 hr onwards, the amino acid levels were always lowest in the double mutants (*vps4Δ*, *atg8Δ*) (*Figure 2A,B*).

To exclude effects contributed by amino acid auxotrophies, the same analysis was performed in a different genetic background with fully prototrophic WT cells and the respective *vps4Δ* and *atg8Δ* single mutants (*Mülleder et al., 2012*). During the first 2 hr of starvation, the amino acid levels initially declined in the prototrophic WT cells, but not as strongly as in auxotrophic strains, and recovered at around 4 hr of starvation, which was dependent on autophagy (*Figure 2—figure supplement 1A,B*). In prototrophic *vps4Δ* mutants, the levels of most amino acids (12) were lower after 2 hr of starvation when compared to WT cells or autophagy mutants (*atg8Δ*), as observed in the auxotrophic strains (*Figure 2A,B*; *Figure 2—figure supplement 1A,B*).

These results showed that the MVB pathway was essential to maintain the levels of most free intracellular amino acids within the first 2 hr of starvation, while autophagy was essential to restore intracellular amino acids later during starvation.

Based on these results we next tested how the MVB pathway would contribute to uphold protein synthesis during starvation. Therefore we measured $^{35}$S-methionine/cysteine incorporation into newly synthesized proteins. Under rich conditions, $^{35}$S-label incorporation was comparable in WT cells, *atg8Δ* and *vps4Δ* single or double mutants (*Figure 2C,D*), although the methionine permease Mup1 was more abundant in ESCRT mutants. Already 1 hr after starvation, $^{35}$S-label incorporation was reduced to 40% in WT cells. During the next 3 hr of starvation, WT cells managed to maintain protein synthesis at this level (*Figure 2C*, lanes 1, 5, 9,13, *Figure 2D*). In the autophagy-deficient *atg8Δ* mutant, $^{35}$S-label incorporation was initially similar to WT cells for up to 2 hr, but began to decline after 4 hr of starvation (*Figure 2C*, lanes 3, 7, 11, 15, *Figure 2D*), which is consistent with the key role of autophagy in amino acid recycling. In ESCRT mutants (*vps4Δ*), protein synthesis declined faster when compared to WT cells or autophagy mutants, which seems consistent with the more rapid decline of intracellular amino acids (*Figure 2C*, lanes 2, 6, 10, 14, *Figure 2D*). *vps4Δ*, *atg8Δ* double mutants showed an additive effect, since even less $^{35}$S-label was incorporated upon starvation compared to the single deletion mutants (*Figure 2C,D*).

Taken together these findings suggest that (i) the MVB pathway is essential to maintain a critical pool of free amino acids for protein synthesis early during starvation. (ii) In the absence of the MVB pathway autophagy can only partially uphold amino acids levels and protein synthesis (iii) The MVB pathway and autophagy cooperate to maintain intracellular amino acids during starvation, potentially in a consecutive manner.

## The ESCRT machinery is not required for the induction of autophagy, the formation of autophagosomes and the delivery of autophagosomes into the vacuole

Recent reports have shown an important role for the ESCRT machinery in higher eukaryotic cells in regulating autophagy at the stage of amphisomes fusing with lysosomes (*Nara et al., 2002*; *Filimonenko et al., 2007*; *Lee et al., 2007*; *Rusten et al., 2007*; *Metcalf and Isaacs, 2010*; *Spitzer et al., 2015*). Therefore we next carefully examined the role of the ESCRT machinery in distinct steps of autophagy in yeast. The induction of autophagy is tightly controlled by TORC1. Under nutrient rich growth conditions, TORC1 was active and its direct targets Sch9 and Atg13 were phosphorylated in WT cells, *vps4Δ* and *atg8Δ* mutants (*Figure 3A,B*, lane 1, 3, 5) (*Kamada et al., 2000*; *Urban et al., 2007*). When autophagy and the MVB pathway were simultaneously disrupted (*vps4Δ*, *atg8Δ*), TORC1 signaling appeared to be reduced under rich growth conditions, but not completely switched off (*Figure 3A*, lane 7). Upon starvation TORC1 signaling was efficiently turned off and the autophagy core component Atg13 was dephosphorylated in all strains, which is a prerequisite for the induction of autophagy (*Figure 3B*) (*Kamada et al., 2000*).

To assess the formation and the delivery of autophagosomes into vacuoles, we followed the transport of GFP-tagged Atg8 using live cell fluorescence microscopy. Upon starvation, GFP-Atg8 was efficiently transported into the lumen of vacuoles in WT cells and *vps4Δ* mutants (*Figure 3C*), indicating that the autophagic machinery was fully operational and independent of the ESCRT machinery. This conclusion was further strengthened using the Pho8Δ60 autophagy-reporter assay (*Noda et al., 1995*). Pho8Δ60 activity increased after 4 hr of starvation in WT and *vps4Δ* mutants, but not in *atg8Δ* cells (*Figure 3D*). Collectively, these results show that in yeast autophagosomes together with their cargo were delivered into the vacuoles of *vps4Δ* mutants in response to starvation, which is consistent with earlier reports (*Reggiori et al., 2004*).

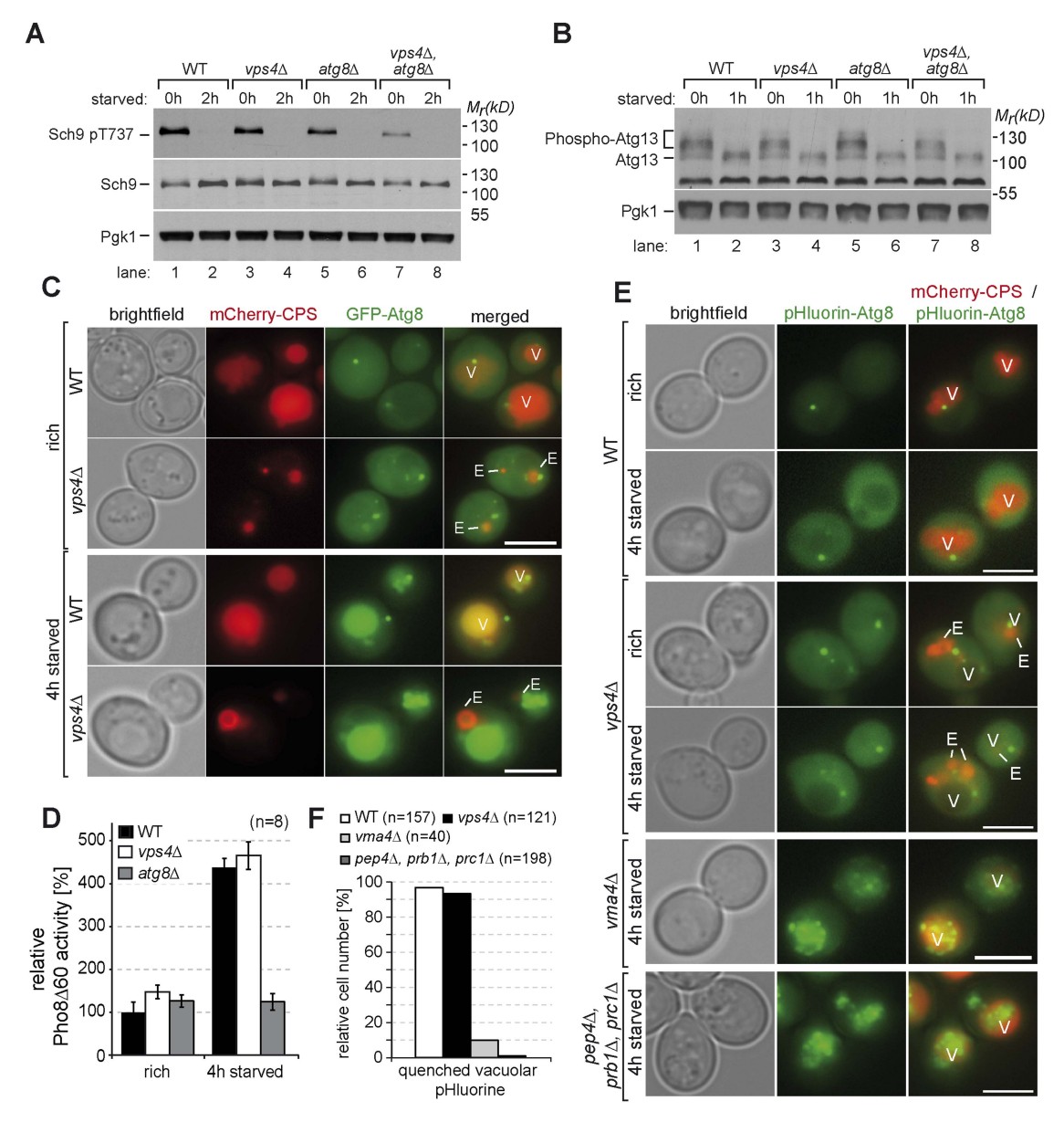

**Figure 3.** Autophagy in ESCRT mutants. (**A**, **B**) SDS-PAGE and western blot analysis of total cell lysates from WT cells and *vps4Δ*, *atg8Δ* single and double mutants grown in rich medium (0 hr) or during starvation using the indicated antibodies. (**C**) Live-cell fluorescence microscopy of WT cells and *vps4Δ* mutants expressing GFP-Atg8 (green) and mCherry-CPS (red) under rich conditions or 4 hr after starvation. (**D**) Pho8Δ60-specific alkaline phosphatase activity was measured in WT, *vps4Δ* and *atg8Δ* cells under rich conditions and after 4 hr of starvation (n = 8, ±SD). WT Pho8Δ60 activity under rich conditions was normalized to 100%. (**E**) Fluorescence microscopy of pHluorin-Atg8 (green) and mCherry-CPS (red) in WT cells and indicated mutants under rich conditions or after 4 hr of starvation. (**F**) Quantification of quenching of vacuolar pHluorin-Atg8 from **E**. (**C**, **E**) (V)acuoles and class (E) compartments. Scale bar = 5 μm.

Next, we analyzed autophagic processes further downstream and examined the lysis of autophagic bodies. This is a prerequisite for the subsequent proteolytic breakdown of autophagic cargo (*Takeshige et al., 1992*; *Yang et al., 2006*) and depends on vacuolar acidification, the catabolic activity of Pep4 and Prb1 and the lipase Atg15 (*Teter et al., 2000*; *Epple et al., 2001*). To determine the breakdown of autophagic bodies in living cells, we generated a functional pHluorin-Atg8 chimera. The fluorescence of pHluorin-Atg8 is detectable at cytosolic pH but not at the lower pH within the vacuole (*Prosser et al., 2010*). In WT cells the fluorescence of pHluorin-Atg8 was efficiently quenched in the lumen of vacuoles

upon starvation (>90% of cells, n = 157). In contrast, in mutants that are either deficient in vacuolar peptidases (*prb1Δ*, *prc1Δ*, *pep4Δ*) or vacuolar acidification (*vma4Δ*) pHluorin-Atg8 was not quenched (<10% of cells, n = 198 and n = 40, respectively) and pHluorin-Atg8 positive vesicular structures were detected inside their vacuoles, suggesting that autophagic bodies were not efficiently lysed (*Figure 3E, F*). In the vast majority of *vps4Δ* mutants (>90% of cells, n = 121), the fluorescence of pHluorin-Atg8 was quenched in the vacuoles similar to WT cells, but occasionally few perivacuolar pHluorin-Atg8 positive structures were observed (<17% of cells, n = 121). Overall, it seemed that autophagic bodies were efficiently lysed in acidified vacuoles of *vps4Δ* mutants (*Figure 3E,F*).

These findings emphasize that in yeast the autophagic machinery, the fusion of autophagosomes with the vacuole per se and the lysis of autophagic bodies is not impaired in ESCRT mutants.

## The MVB pathway is required for early proteome remodeling during starvation

Next we determined in detail how the MVB pathway would contribute to the starvation program of yeast. Therefore we measured how the proteome of WT (*vps4Δ* complemented with *VPS4*) cells changed within the first 3 hr of starvation using stable isotope labeling with amino acids in cell culture (SILAC) (*de Godoy et al., 2008*). WT cells were grown under rich conditions with heavy $^{13}C_6^{15}N_2$-lysine or light $^{12}C_6^{14}N_2$-lysine, and light cells were subsequently starved for 3 hr. Equal cell numbers were mixed prior to lysis and mass spectrometry (MS) analysis (*Figure 4—figure supplement 1A*, upper panel). In total 2941 proteins were quantified (peptide count ≥ 2), comprising 58% of the characterized yeast ORFs (*Figure 4A*, *Figure 4—figure supplement 1A*, upper panel, *Supplementary file 1*). In this early phase of starvation, the yeast proteome already underwent extensive remodeling and 264 proteins significantly changed in abundance (MaxQuant significance B) (*Cox and Mann, 2008*). 101 proteins were significantly down- and 163 proteins significantly up-regulated.

To identify cellular components and biological processes that were regulated early during starvation, we determined the enrichment of specific Gene Ontology (GO) terms (*Harris et al., 2004*) (*Figure 4A*, *Figure 4—figure supplement 1B*, *Supplementary file 2*). Amino acid- and carbohydrate-metabolic pathways were up-regulated to meet the demands of starving cells. Most vacuolar hydrolases were also up-regulated in response to nutrient limitation, indicating that starving cells enhanced the catabolic activity and thus the recycling capacity of their vacuoles. Components of the autophagic machinery, including Atg1 and Atg8, were also up-regulated (*Figure 4A*). In contrast, cell wall components, proteins required for ribosome biogenesis and cell cycle/division were down-regulated (*Figure 4A*, *Figure 4—figure supplement 1B*). This reflects known cellular responses to starvation, such as reduced cell growth and protein synthesis, exit from mitosis and entry into quiescence.

Strikingly, the GO analysis also revealed that plasma membrane proteins were among the most frequently down-regulated cellular components during starvation. Not only Mup1, but many other plasma membrane proteins, including diverse high affinity nutrient permeases required for the transport of sugars (Itr1, Hxt2, 3), nucleobases (Fur4), amino acids (Bap3, Gnp1, Tat1, Can1) and ammonium (Ato3) but also the G-protein coupled receptor Ste2 were down-regulated (*Figure 4A*) and degraded via the MVB pathway (*Figure 4—figure supplement 2A*). This comprehensive starvation-induced remodeling of the plasma membrane was highly selective. The protein levels of the plasma membrane H$^+$-ATPase, Pma1, were not dramatically altered. The majority of Pma1 remained at the cell surface and only a small portion was delivered into the vacuole (*Figure 4—figure supplement 2A*). Moreover, the general amino acid permease Gap1 and the ammonium permease Mep2 were strongly up-regulated and the majority of Gap1 was retained at the plasma membrane (*Figure 4A*, *Figure 4—figure supplement 2B,C*).

To define how the MVB pathway contributed to proteome remodeling, the same proteomic experiment was performed with an isogenic ESCRT mutant (*vps4Δ*) (*Figure 4—figure supplement 1A* lower panel, *Supplementary file 3*). To compare starvation-induced proteome remodeling in WT cells and *vps4Δ* mutants, we restricted our analysis to 2694 proteins that were reliably quantified in both strains under rich and starvation conditions (*Figure 4B*, *Supplementary file 3*). Correlation analysis (R = 0.82, p < 1e-16) revealed that WT cells and *vps4Δ* mutants up- or down-regulated similar proteins in response to starvation, suggesting that ESCRT mutants are not generally deficient in inducing a starvation response (*Figure 4B*). However, data correlation analysis indicated that a majority of proteins in *vps4Δ* mutants showed less pronounced changes during starvation (*Figure 4B*). To address

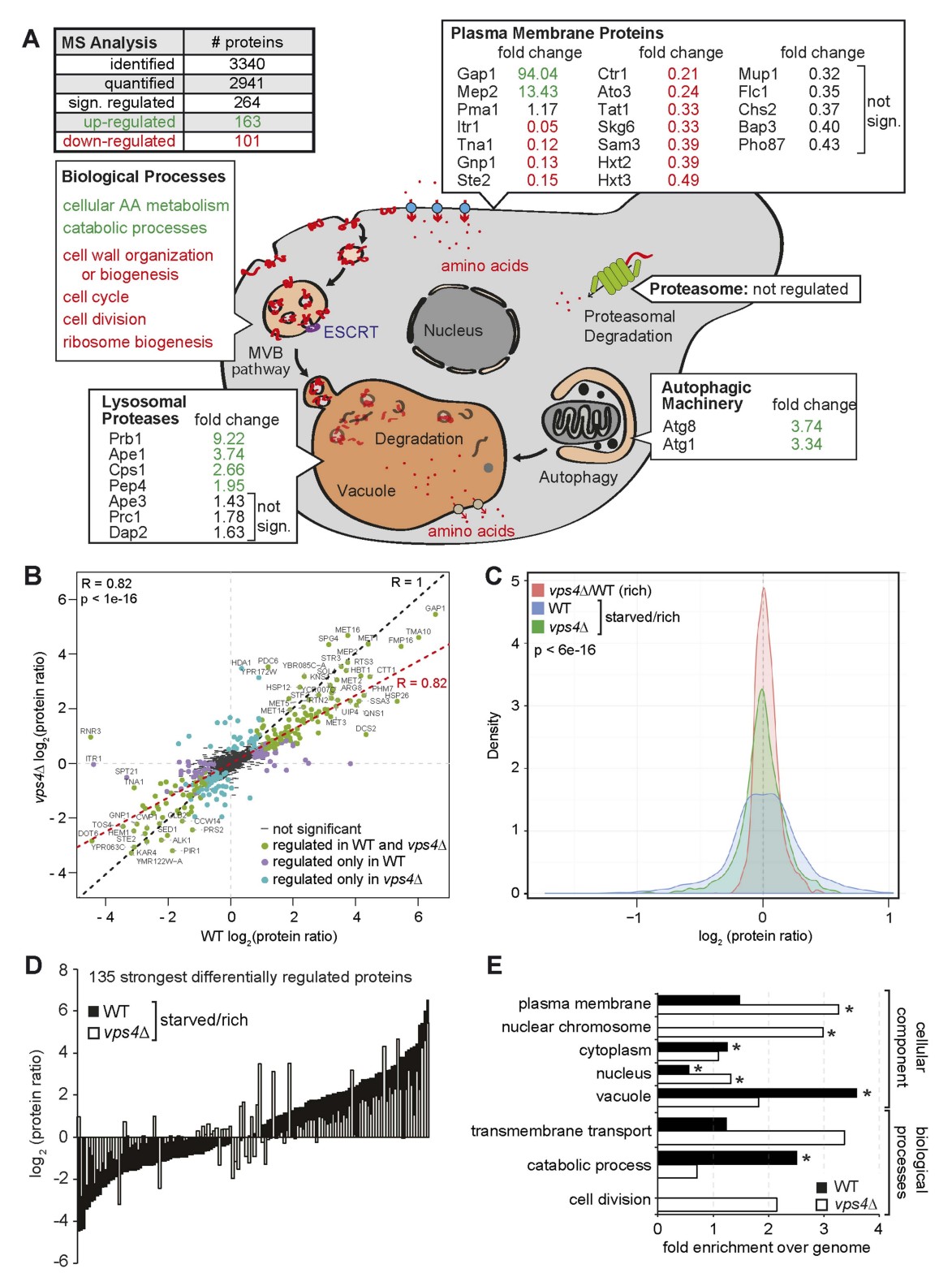

**Figure 4**. The MVB is required for starvation induced proteome remodeling. (**A**) Schematic presentation of proteome remodeling in WT cells during starvation. Starvation induced changes in protein levels were measured using SILAC based quantitative proteomics (see also **Figure 4—figure supplement 1A,B**, **Supplementary files 1, 2**). The major changes in WT cells under starvation as indicated by Gene Ontology (GO) analysis of significantly changed proteins are shown. Green: up-regulated; red: down-regulated under starvation. (**B**) Correlation of changes in protein abundance in

*Figure 4. continued on next page*

*Figure 4. Continued*

WT cells and *vps4Δ* mutants during starvation (see also *Figure 4—figure supplement 1A*, *Supplementary file 3*). WT and *vps4Δ* mutant protein ratios (log2 [starved/rich]) are plotted against each other. Green: significantly regulated in both datasets; blue: significantly regulated only in *vps4Δ*; purple: significantly regulated only in WT. Grey: not significantly regulated. (C) Density plot showing log2-transformed protein ratio distributions in the three quantitative proteome datasets. The significant protein changes are excluded. Blue: WT (starved)/WT(rich); green: *vps4Δ*(starved)/*vps4Δ*(rich); red: *vps4Δ* (rich)/WT(rich) (see also *Figure 4—figure supplement 1*, *Supplementary file 4*). p-value according to Kolmogorov–Smirnov $<6 \times 10^{-16}$ (D) The 135 significantly differentially regulated proteins during starvation between WT cells (black bars) and *vps4Δ* mutants (white bars) (see also *Supplementary file 5*). (E) Enrichment of GO terms in 135 significantly differentially regulated proteins (from D). Data are represented as fold-enrichment over whole genome frequency (see also *Supplementary file 6*). * significantly represented GO terms.

The following figure supplements are available for figure 4:

**Figure supplement 1**. Quantitative proteomics and GO analysis.

**Figure supplement 2**. Starvation induced endocytosis.

this observation over the entire datasets, we calculated the frequency by which changes in protein abundance occurred in WT cells or *vps4Δ* mutants. This analysis showed that the distribution of protein ratios was broader in WT cells (blue curve) than in the *vps4Δ* mutants (green curve) (*Figure 4C*, p < 6e-16). Hence in WT cells more proteins were stronger up- or down-regulated under starvation compared to ESCRT mutants, where changes in protein levels were less pronounced.

An additional direct quantitative analysis of the proteomes of WT cells (labeled with heavy $^{13}C_6^{15}N_2$-lysine) and *vps4Δ* mutants growing under rich conditions (*Figure 4—figure supplement 1C,D*, *Supplementary file 4*) showed the narrowest ratio distribution (red curve) (*Figure 4C*). These results indicate that the MVB pathway has a small and selective effect on the proteome of cells growing under rich conditions, but becomes critical to support proteome remodeling once extra-cellular amino acids become limiting.

To pinpoint processes that were particularly dependent on the MVB pathway during starvation, we identified 135 proteins that showed the most significant differences in changes of protein abundance between WT cells and *vps4Δ* mutants (*Figure 4D*, *Supplementary file 5*). From this analysis it became additionally evident that most proteins were (with few exceptions) stronger up- or down-regulated in WT cells (black bars) than in *vps4Δ* mutants (white bars). GO analysis of these 135 proteins identified three processes that were primarily differentially regulated between *vps4Δ* mutants and WT cells upon starvation (*Figure 4E*, *Supplementary file 6*). Based on this analysis we conclude that in the first 3 hr of starvation the MVB pathway is particularly required (i) for the degradation of plasma-membrane proteins, as expected, which probably helps to maintain intracellular amino acid levels; (ii) to increase the protein levels of vacuolar hydrolases and thereby enhance the catabolic processes in vacuoles and also (iii) to down-regulate proteins that control the cell division cycle.

## Boosting vacuolar catabolism early during starvation depends on the de novo synthesis of vacuolar hydrolases, which requires the MVB pathway

Our quantitative proteomic analysis indicated that most vacuolar proteases, in particular Prb1, but also Ape1, Cps1 and Pep4, were up-regulated during the first 3 hr of starvation in WT cells (*Figures 4A, 5A*). Also other types of vacuolar hydrolytic enzymes were induced during that time, like the alpha-mannosidase Ams1 or the ribonuclease Rny1. To directly assess how the catabolic activity of vacuoles changed during starvation, we measured the enzymatic activity of vacuolar alkaline phosphatase, Pho8. The transmembrane protein Pho8 is delivered to the vacuole via the AP-3 pathway, which functions independently of the ESCRT machinery (*Cowles et al., 1997*). Pho8 activity requires proteolytic maturation of Pho8 by Pep4 on the C-terminus. An additional uncharacterized endoproteolytic activity further cleaves mPho8 to yield a soluble sPho8 inside the vacuole that can be specifically measured (*Figure 5B,C*) (*Song, 2006*). In a yeast mutant that was deficient for three major vacuolar peptidases (*pep4Δ, prb1Δ, prc1Δ*), Pho8 was not matured (*Figure 5B*, lane 1) and Pho8 activity was not detected (*Figure 5C*). In *vps4Δ* mutants the Pep4-dependent maturation to mPho8 was not impaired under rich growth conditions or starvation (*Figure 5B*, lane 3–5). In all cells sPho8 levels were low under rich conditions, which corresponded

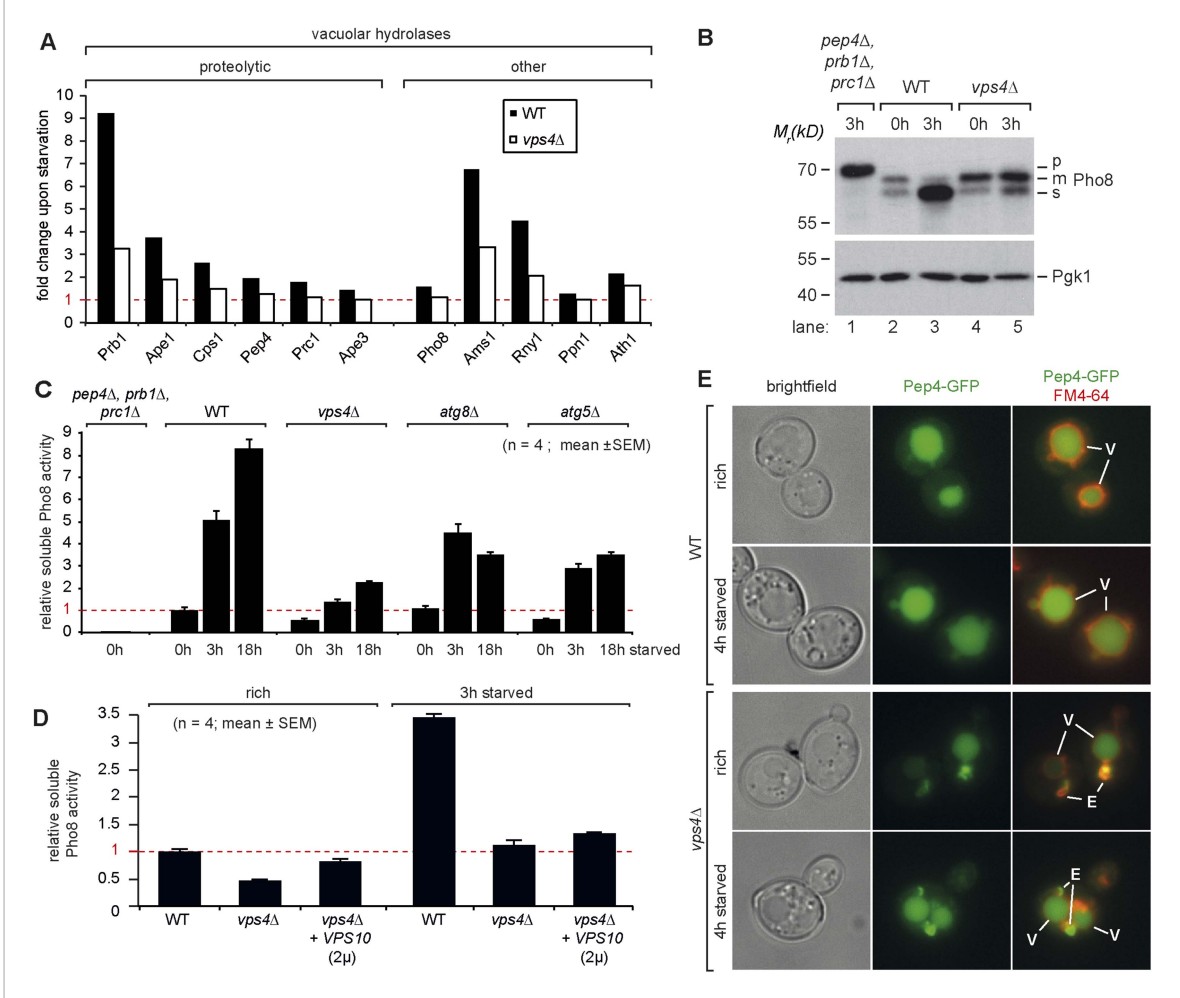

**Figure 5**. Boosting the catabolic activity of vacuoles during starvation requires the MVB pathway. (**A**) Starvation-induced changes in the protein levels of various vacuolar hydrolases based on SILAC data in WT (black) and *vps4Δ* mutants (white). (**B**) Indicated yeast strains were grown in rich medium (0 hr) or starved for 3 hr. Cell lysates were subjected to SDS-PAGE and western blot analysis with the indicated antibodies. p, precursor form; m, mature form; s, soluble form. (**C**, **D**) Soluble Pho8 (sPho8) activity in rich medium and upon starvation of the indicated strains (mean ± SEM, n = 4). (**E**) Fluorescence microscopy of Pep4-GFP (green) in WT cells and *vps4Δ* mutants growing under rich or starvation conditions. (V)acuoles (FM4-64, red) and class (E) compartments. Scale bar = 5 μm. *vps10Δ + VPS10 (2 μ)* cells were used as the isogenic WT control in (**D**) and (**E**).

The following figure supplement is available for figure 5:

**Figure supplement 1**. Boosting the catabolic activity of vacuoles requires membrane protein degradation via the MVB pathway.

to low Pho8 activity (*Figure 5B,C*). Within the first 3 hr of starvation, sPho8 activity increased at least fivefold in WT cells (*Figure 5C*), consistent with the de novo synthesis of vacuolar proteases resulting in increased endoproteolytic activity generating sPho8 (*Figure 5B* lane 3). In the following 15 hr of starvation Pho8 activity only doubled (*Figure 5C*). Hence the major boost for the catabolic activity of vacuoles occurs during the first 3 hr of starvation. In two different mutants that block autophagy (*atg8Δ* and *atg5Δ* mutants), sPho8 activity still increased 3–5 fold during the first 3 hr of starvation similar to WT cells, but failed to increase further upon extended starvation (*Figure 5C*), suggesting that autophagy was not required to boost the catabolic activity of vacuoles early during starvation.

In *vps4Δ* mutants growing under rich conditions, sPho8 protein levels were slightly lower (*Figure 5B*, lane 4), consistent with lower Pho8 activity (*Figure 5C*). Upon starvation sPho8 activity merely doubled in *vps4Δ* mutants during first hours of starvation and never reached levels comparable

to WT cells or autophagy mutants (*Figure 5C*). ESCRT-I (*vps23Δ*) or ESCRT-II (*vps36Δ*) mutants were also severely impaired in increasing the catabolic activity of their vacuoles during starvation (*Figure 5—figure supplement 1A*).

SILAC based quantification of vacuolar hydrolases in starving *vps4Δ* mutants revealed that the protein levels of most vacuolar hydrolases including Prb1, Ape1, Cps1 and Pep4 were only marginally induced (*Figure 5A*). Therefore mPho8 was not efficiently cleaved to sPho8 in *vps4Δ* mutants (*Figure 5B* lane 5) and the catabolic activity of vacuoles in ESCRT mutants failed to increase early during starvation (*Figure 5C*). The inability of ESCRT mutants to increase the protein levels of vacuolar proteases appears to culminate in a failure to boost the catabolic activity of vacuoles early during starvation. This is best explained by the central role of the MVB pathway in maintaining intracellular amino acid levels for protein synthesis early during starvation.

## Boosting the catabolic activity of vacuoles requires membrane protein degradation via the MVB pathway

ESCRT mutants also have a minor sorting defect for soluble vacuolar hydrolases, which are aberrantly secreted, mainly because Vps10, the sorting receptor for multiple vacuolar hydrolases (e.g.: Pep4 and Prc1) recycles less efficiently between endosomes and the golgi (*Bankaitis et al., 1986*; *Rothman and Stevens, 1986*). Hence, severe mis-sorting and strong secretion of vacuolar hydrolases might alternatively explain the failure of ESCRT mutants to recycle amino acids, maintain protein synthesis and boost the catabolic activity of vacuoles.

Therefore we next analyzed the extent to which the mis-sorting of vacuolar hydrolases would contribute to the here describe phenotypes of ESCRT mutant. First, we determined the subcellular localization of the master protease Pep4-GFP and Prc1/CPY-RFP in ESCRT mutants under rich growth or starvation using live cell fluorescence microscopy. Pep4-GFP localized to the class E compartment in ESCRT mutants, but a large fraction of Pep4-GFP was also delivered into the lumen of the vacuole (*Figure 5E*). Similar results were obtained for a construct containing the vacuolar sorting signal of Prc1/CPY fused to RFP (*Figure 5—figure supplement 1B*). Hence, a considerable fraction of vacuolar hydrolases still arrived in the lumen of the vacuole where they fully matured and were active (*Figures 3E, 5*). Notably, ESCRT mutants, unlike other endo-lysosomal trafficking complexes including HOPS, CORVET or retromer, were never identified as *pep* mutants (*Jones, 1977*) because they displayed relatively minor mis-sorting of vacuolar hydrolases (*Bankaitis et al., 1986*; *Rothman and Stevens, 1986*), which kept their vacuoles catabolically active.

Earlier reports suggested that the overexpression of Vps10 can selectively rescue the partial mis-sorting of vacuolar hydrolases (CPY/Prc1, Pep4) in *vps4Δ* mutants but not the degradation of membrane proteins (*Babst et al., 1998*). As expected, Mup1-GFP still localized to class E compartments and was not delivered into vacuoles in *vps4Δ* cells overexpressing Vps10 (*Figure 5—figure supplement 1B*). Yet, Vps10 overexpression alleviated mis-sorting of vacuolar enzymes (*Figure 5—figure supplement 1C*) and thus increased the catabolic activity of vacuoles in *vps4Δ* mutants to 80% of WT levels under rich growth conditions (*Figure 5D*). Despite this restoration of vacuolar catabolic activity prior to starvation, overexpression of Vps10 failed to rescue intracellular amino acid levels (*Figure 5—figure supplement 1D*) and protein synthesis (*Figure 5—figure supplement 1E,F*) in *vps4Δ* mutants throughout starvation.

It seems that the cellular defects of ESCRT mutants during starvation can be mostly attributed to their inability to degrade membrane proteins. *vps4Δ* mutants but also *vps4Δ* mutants overexpressing Vps10 could not efficiently up-regulate the de novo synthesis of vacuolar hydrolases in time. Therefore the catabolic activity of vacuoles remained low during starvation in *vps4Δ* mutants overexpressing Vps10 (*Figure 5D*). This is further emphasized by the impairment of ESCRT mutants to efficiently increase the protein levels of two other hydrolases, Ape1 and Ams1, during starvation (*Figure 5A*). Both Ape1 and Ams1 are delivered to vacuoles via the cvt-pathway and hence will not be secreted or mis-sorted in ESCRT mutants.

All of these findings are consistent with the idea that selective membrane protein degradation via the MVB pathway, rather than the mere sorting of vacuolar hydrolases, is essential to maintain sufficient free intracellular amino acids to uphold protein synthesis for proteome remodeling early during starvation. It thereby contributes essentially to the de novo synthesis of vacuolar hydrolases to concomitantly boost of the catabolic activity of vacuoles.

# Boosting the catabolic activity of vacuoles early during starvation is essential for the efficient degradation of autophagic cargo

Next we tested if the MVB-dependent de novo synthesis of vacuolar hydrolases and the subsequent boost in hydrolytic activity was also required to break down and recycle autophagic cargo.

Our proteomic studies comparing WT cells and *vps4Δ* mutants growing under rich conditions indicated that Atg8 protein levels were increased in *vps4Δ* mutants when compared to WT cells (*Figure 4—figure supplement 1C*, *Supplementary file 4*). This was also confirmed by western blot analysis (*Figure 6A*, lane 5, *Figure 4—figure supplement 1D*). Despite the efficient transport of GFP-Atg8 into vacuoles and lysis of autophagic bodies (*Figure 3C,E*), the release of free GFP from Atg8, which depends on efficient vacuolar proteolysis, was delayed in *vps4Δ* mutants, but not completely blocked (*Figure 6A*).

Similarly, already under rich growth conditions *vps4Δ* mutants had higher protein levels of mature mApe1 and immature pApe1 (*Figure 6B*, lane 1, 2), which is delivered to vacuoles via the cvt-pathway

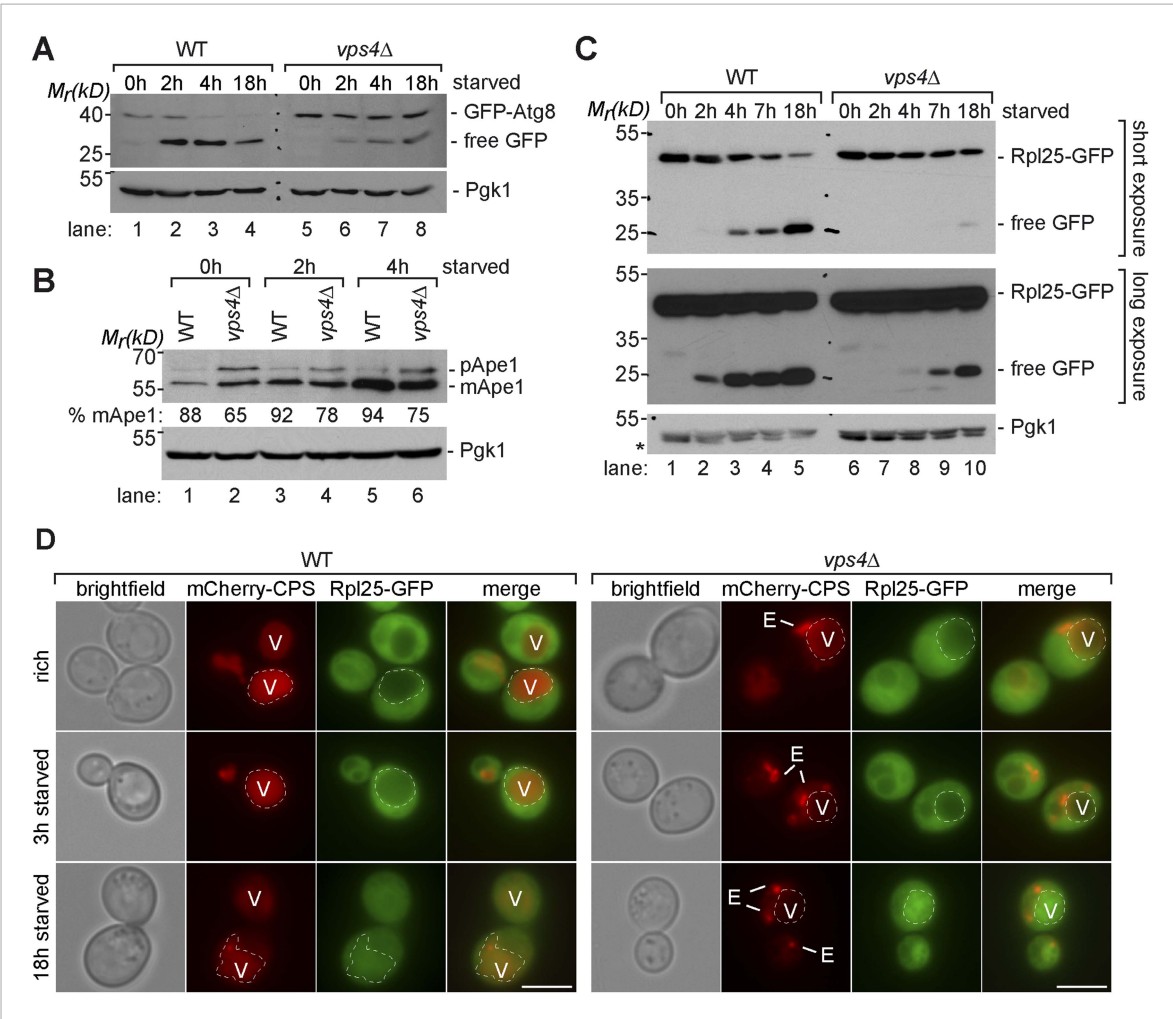

**Figure 6**. Boosting the catabolic activity of vacuoles is essential for the efficient degradation of autophagic cargo. (A, B, C) SDS-PAGE and western blot analysis of total cell lysates from WT cells and *vps4Δ* mutants grown in rich medium or during starvation using the indicated antibodies. p(ro)Ape1, m(ature)Ape1. *residual anti-GFP signal after re-probing the membrane with anti-Pgk1 antibody. (D) Fluorescence microscopy of Rpl25-GFP (green) and mCherry-CPS (red) in WT cells and *vps4Δ* mutants under rich conditions or after starvation. Dashed lines indicate the vacuolar membrane. (V)acuoles and class (E) compartments. Scale bar = 5 µm.

The following figure supplement is available for figure 6:

**Figure supplement 1**. Proteolytic processing of autophagic cargo.

(*Klionsky et al., 1992*). Starving WT cells strongly induced the expression of Ape1 and increased its autophagy-dependent transport to the vacuole (*Baba et al., 1997*), indicated by its efficient proteolytic processing (*Figure 6B*, lanes 3, 5, *Figure 5A*). In starving *vps4Δ* mutants, mApe1 also increased, but not as strongly as in WT cells, which is consistent with our quantitative proteomic data (*Figure 5A*, *Supplementary file 3*). In *vps4Δ* mutants we always detected more pApe1, indicating a delay in proteolytic maturation (*Figure 6B*, lane 6). In contrast when *PEP4* or *PRB1* were deleted, the release of free GFP from GFP-Atg8 and Ape1 maturation were fully blocked, further confirming that the catabolic activity of ESCRT mutants vacuoles was by no means completely defective (*Figure 6—figure supplement 1A*).

Additionally, the autophagy-dependent proteolytic processing of the ribosomal subunit Rpl25-GFP was delayed in *vps4Δ* mutants (*Figure 6C*, lane 7–10), although it was transported into the vacuoles of *vps4Δ* mutants (*Figure 6D*). The degradation of autophagy cargo during starvation was also not restored in *vps4Δ* mutants by Vps10 overexpression (*Figure 6—figure supplement 1B*).

These results provide further evidence that amino acid recycling through the selective degradation of membrane proteins via the MVB pathway is required to boost the vacuolar catabolic activity early during starvation, mainly by maintaining protein synthesis and therefore promoting the up-regulation of vacuolar hydrolases. This order of events primes vacuoles for the efficient degradation of autophagic cargo.

## The coordinated function of the MVB pathway and autophagy is required to enter quiescence upon starvation

Starving cells have to exit proliferation and enter quiescence to survive this stress condition. In yeast, starvation results in a stable $G_1/G_0$ arrest, which can be scored by the increase of unbudded cells. Starving WT cells grew slowly, but still doubled their optical density (*Figure 7A*) and the majority (>80%) no longer displayed budding daughter cells (*Figure 7B*). Hence WT cells managed to complete a final cell division cycle and efficiently arrested in $G_1/G_0$ during starvation. Autophagy mutants (*atg8Δ* or *atg5Δ*) grew slower when compared to WT cells upon starvation and failed to complete cell division and thus could not enter $G_1/G_0$ arrest (*Figure 7A,B*), consistent with recent reports showing that autophagy is essential to overcome a Swe1-dependent checkpoint mechanism (*Matsui et al., 2013*; *An et al., 2014*). Our proteomic analysis indicated that cell cycle regulatory proteins (including Swe1) were less efficiently down-regulated during starvation in *vps4Δ* mutants (*Figure 4E*, *Supplementary file 3*). Strikingly, when *vps4Δ* mutants were subjected to starvation, their growth slowed down prior to the growth of autophagy mutants (*Figure 7A*). ESCRT-I (*vps23Δ*), ESCRT-II (*vps36Δ*), ESCRT-III (*snf7Δ*) and *vps4Δ* mutants failed to complete their final round of cell division and hence could not enter a $G_1/G_0$ arrest (*Figure 7B*). When the partial sorting defect of vacuolar hydrolases of ESCRT mutants was rescued by Vps10 overexpression, *vps4Δ* mutants still stopped to grow and failed to enter quiescence during starvation (*Figure 7—figure supplement 1A,B*). A mutant deficient for three vacuolar peptidases (*pep4Δ*, *prb1Δ*, *prc1Δ*) (*Figure 7B*) also failed to complete cell division and could not enter $G_1/G_0$ arrest (*Figure 7B*). In this mutant the sequestration of cargo into MVBs or autophagosomes and their transport into vacuoles is not affected (*Figure 3E*). Hence these results implicate that proteolytic degradation of MVB cargo and autophagic cargo inside the vacuoles in general, rather than removal/sequestration of a specific factor, is essential to complete a final cell division cycle and enter quiescence during starvation.

Finally, survival of long-term starvation was assessed by placing equal amounts of cells from starving cultures onto rich medium plates. WT cells barely lost viability even when they were starved for 13 days. In contrast, both *vps4Δ* and *atg8Δ* mutants gradually lost viability over time, and the *vps4Δ*, *atg8Δ* double mutants showed severe synthetic survival defects (*Figure 7C*).

In summary these results show that starvation-induced endocytosis and the subsequent selective degradation of these membrane proteins via the MVB pathway maintains intracellular amino acids pools that are required for the synthesis of new proteins early during starvation. This includes the de novo synthesis of vacuolar hydrolases, which is essential to boost the catabolic activity of vacuoles (*Figure 7D*). This order of events primes vacuoles for the efficient degradation of bulk cytoplasm by autophagy, enables the continuous recycling of nutrients to maintain cellular homeostasis during extended periods of starvation and thereby promotes a stable cell cycle arrest that ensures cell survival.

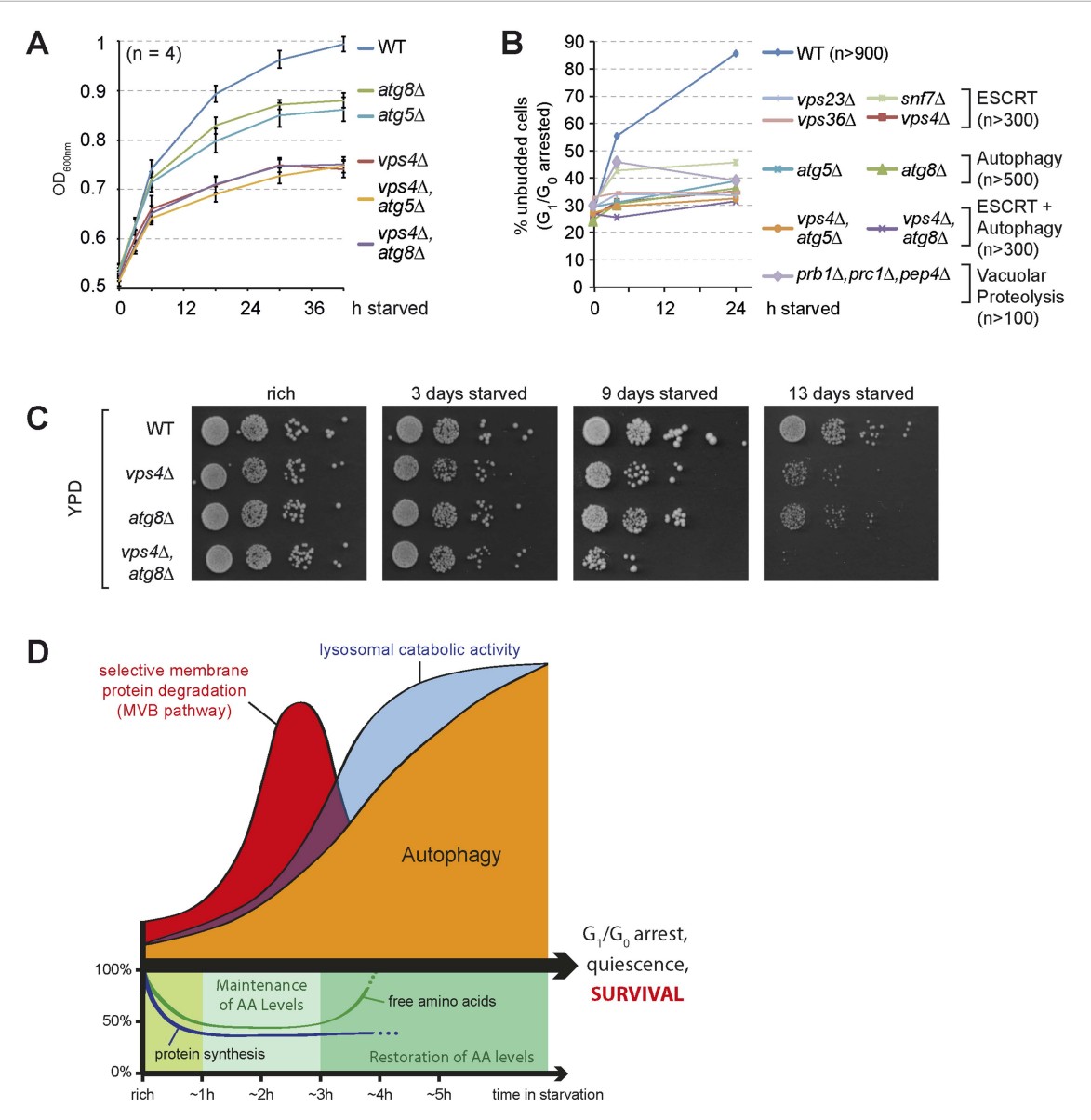

**Figure 7**. The coordinated function of the MVB pathway and autophagy is required to enter quiescence upon starvation. (**A**) Growth of WT cells and the indicated mutants after shift from logarithmic growth in rich medium (0 hr) to starvation measured with $OD_{600nm}$. Mean ± SEM, n = 4. (**B**) Quantification of unbudded cells ($G_1/G_0$ arrested) under rich conditions (0 hr) or after indicated time of starvation. (**C**) Cells were starved for the indicated times and equal amounts of cells in serial dilutions were placed on rich medium (YPD). (**D**) Model representing the coordinated action of lysosomal protein degradation pathways for amino acid maintainance and recycling as well as protein synthesis during starvation.

The following figure supplement is available for figure 7:

**Figure supplement 1**. Cell growth and entry into quiescence upon starvation.

## Discussion

Here we show that eukaryotic cells utilize a catabolic cascade of selective and non-selective degradation pathways to ensure cell survival upon starvation (*Figure 7D*). Immediately upon nutrient limitation, the degradation of cytoplasmic proteins by the UPS acutely supplies amino acids to maintain protein synthesis (*Vabulas and Hartl, 2005*). Starvation also inactivates TORC1 signaling, which probably simultaneously stimulates autophagy as well as starvation-induced endocytosis.

Once TORC1 signaling is turned off, Atg13 is quickly dephosphorylated, which allows the formation of the Atg1/Ulk1 kinase complex to induce autophagy (*Kamada et al., 2000*, *2010*).

The TORC1/Npr1 signaling axis also controls at least in part starvation-induced endocytosis, by regulating different arrestin-like adaptors (ARTs) for Rsp5, the major ubiquitin ligase required for endocytosis in yeast. Thereby TORC1 signaling orchestrates the selective remodeling of the plasma membrane proteome under different growth conditions and additionally could increase protein turnover via the MVB pathway (*Schmidt et al., 1998*; *Beck et al., 1999*; *Léon et al., 2008*; *MacGurn et al., 2011*; *Jones et al., 2012*; *Boeckstaens et al., 2014*; *Crapeau et al., 2014*). Our results demonstrate for the first time that starvation induces a massive but still selective remodeling of the plasma membrane proteome. At least 18 different integral plasma membrane proteins undergo starvation-induced endocytosis including different amino acid permeases, sugar transporters and the G-protein coupled receptor, Ste2. Within a few hours these membrane proteins are transported via the MVB pathway into vacuoles for degradation. How TORC1 signaling could control Rsp5-dependent cargo specificity and the selective ubiquitinylation of many different cargoes at the same time is currently not clear.

Our results further demonstrate that the ESCRT-dependent degradation of these membrane proteins in vacuoles is essential to maintain a critical pool of free amino acids for protein synthesis. This enables the de novo synthesis of vacuolar hydrolases that are required to boost the catabolic activity of vacuoles (*Figure 7D*). Thereby, the MVB pathway ensures that the catabolic activity of the vacuole is up-regulated in time to allow subsequent, efficient degradation of autophagic cargo. Only when these conditions are met, autophagy considerably contributes to amino acid recycling during extended starvation to restore intracellular amino acid levels (*Komatsu et al., 2005*; *Onodera and Ohsumi, 2005*). Since the MVB pathway is also in part required for the proper targeting of hydrolases into the vacuole, it might even fulfill a dual function. Yet, our results emphasize that a significant fraction of vacuolar hydrolases are sorted into the lumen of vacuoles in ESCRT mutants, where they mature and become active. Furthermore, if the observed phenotypes in ESCRT mutants would be caused solely by mis-sorting of vacuolar hydrolases, this would lead to a delay in autophagy-mediated amino acid recycling. In this case their defects during starvation should be similar or even weaker when compared to autophagy mutants. The partial sorting defects for vacuolar hydrolases are thus not consistent with the stronger defects of ESCRT mutants in maintaining amino acid levels, protein synthesis and cell growth early during nutrient limitation. Moreover it was possible to restore the catabolic activity of vacuoles in ESCRT mutants under rich growth conditions, while this was no longer possible during starvation. Hence it seems that mainly the selective degradation of membrane proteins as a source for amino acids, rather than just the sorting of vacuolar hydrolases, contributes to the key role of the MVB pathway during starvation. At the moment it is not clear why the MVB pathway appears to be more critical than autophagy to maintain intracellular amino acids levels early during starvation. We speculate that membrane proteins that have been selected for degradation via the MVB are for the most part not re-synthesized, while non-selective autophagy will inevitably also capture and degrade proteins that are still needed and thus have to be replaced. Hence, selective protein degradation may at least initially provide a bigger added value for the free intracellular amino acid pool as compared to the non-selective degradation of bulk cytoplasm by autophagy. The ESCRT machinery has recently been described to contribute to selective micro-autophagy on late endosomes in mammalian cells (*Sahu et al., 2011*). Hence, ESCRT-dependent catabolic pathways may not be limited to membrane proteins, although our study did not experimentally address this possibility.

Ultimately our results suggest that only the coordinated action of the MVB pathway and autophagy provides sufficient intracellular recycling capacity during starvation to allow efficient mitotic exit and entry into a stable $G_1/G_0$ quiescent state and thereby ensures cell survival (*Figure 7D*). The underlying mechanisms are not fully understood but probably require amino acid recycling from the vacuole to the cytoplasm via vacuolar amino acid permeases. Alternatively, the evolutionary conserved EGO/LAMTOR/Ragulator complex in conjugation with a lysosomal amino acid sensor (*Dubouloz et al., 2005*; *Sancak et al., 2010*; *Zoncu et al., 2011a*; *Rebsamen et al., 2015*; *Wang et al., 2015*) could somehow measure free amino acids in the lumen of lysosomes and transiently re-activate TORC1 to complete the final cell division cycle prior to entry into quiescence (*Matsui et al., 2013*; *An et al., 2014*).

In *Drosophila* and human cells, loss of the ESCRT machinery interferes with a late step in autophagy, namely the fusion of amphisomes with lysosomes (*Nara et al., 2002*; *Filimonenko et al., 2007*;

*Rusten et al., 2007*; *Lee and Gao, 2009*). Amphisomes are acidic pre-lysosomal hybrid organelles of MVBs/late endosomes and autophagosomes (*Stromhaug and Seglen, 1993*), Thus, our findings that a functional MVB pathway is key to boost the catabolic activity may not be restricted to lysosomes but may also include MVB-derived amphisomes and thereby ensure the efficient degradation of autophagic cargo in higher eukaryotes. In yeast autophagosomes appear to fuse directly with vacuoles and amphisomes have not been described. Consistently, we and others demonstrated that loss of ESCRT function does not significantly impair the autophagic machinery itself or the delivery of autophagic cargo into lysosomes in yeast or *Caenorhabditis elegans* (*Reggiori et al., 2004*; *Djeddi et al., 2012*; *Jones et al., 2012*).

Our results show that the MVB pathway takes a central role in cellular homeostasis to preserve and redistribute biomass that can be used to maintain cell growth under nutrient limitation. It is tempting to speculate that during transient nutrient fluctuations or changes in metabolism a more stepwise activation of this catabolic cascade with initial selective protein degradation via UPS and the MVB pathway might help to delay massive non-selective breakdown of cytoplasm via autophagy, at least for some time. This might provide a safety mechanism to protect cells from the immediate need for non-selective protein degradation. While the ESCRT machinery has acquired additional roles in diverse biological processes in higher eukaryotes, we propose that the central role of the MVB pathway in the catabolic cascade of eukaryotic cells during starvation is evolutionary conserved.

## Material and methods

### Yeast strains and growth conditions

All experiments were performed with SEY6210 yeast strains, except for Pho8Δ60-expressing strains, strains used in *Figure 1C* and *Figure 6—figure supplement 1A* and prototrophic strains (*Figure 2—figure supplement 1A,B*), which were derived from BY4741 or BY4742. For growth under rich conditions, cells were incubated in YNB synthetic medium supplemented with amino acids/nucleobases (Ade, Arg, Lys, Thr, Tyr plus Ura, Trp, Leu or His when required for auxotrophic strains) and 2% glucose, at 26°C, except for *Figure 3B* (YPD). For starvation experiments, cells were kept at mid-log phase for 24 hr before they were twice washed with and resuspended in YNB with 2% glucose but w/o amino acids and $(NH_4)_2SO_4$. For growth on agar plates, yeast cells were diluted to $OD_{600nm}$ = 0.05 and spotted in 10× dilutions on YPD or YNB plates. Protein synthesis was inhibited by treatment with cycloheximide (Sigma Aldrich, Austria, 50 μg/ml) and proteasomal activity was blocked by MG132 (Sigma Aldrich, 50 μM).

### Yeast strains, plasmids and cloning

Genetic modifications were done by PCR and/or homologous recombination using standard techniques. Where applicable, tags were introduced at the C-terminus to preserve the endogenous promoter sequences. Plasmid-expressed genes including their endogenous promoters were amplified from yeast genomic DNA into centromeric vectors (pRS series). All constructs were analyzed by DNA-sequencing and transformed into yeast cells using standard techniques. Yeast strains and plasmids used in this study are listed in *Supplementary file 7* and primer in *Supplementary file 8*.

### Sample preparation for MS

For quantitative proteomics yeast cells were grown in complex synthetic medium (CSM -His, -Arg, -Lys, complemented with Arginine and Lysine, SunriseScience Products, San Diego, CA) and kept at mid-log phase for 24 hr. Subsequently, cells were washed with their corresponding labeling medium and then used to inoculate 500 ml of labeling medium. Cells were kept in log phase for 10 generations with either, heavy $^{13}C_6^{15}N_2$-L-Lysine or unlabeled $^{12}C_6^{14}N_2$-L-Lysine. For starvation experiments, logarithmically growing cells from unlabeled medium were washed twice with starvation medium and incubated for 3 hr in starvation medium. Cells were harvested by centrifugation. Labeled and unlabeled cells were mixed in a 1:1 ratio according to their $OD_{600nm}$. Cell were mechanically disrupted with glass beads at 4°C in PBS containing protease inhibitors (Aprotinin 10 μg/ml; Pepstatin 1 μg/ml, Leupeptin 10 μg/ml, Pefablock SC 100 μg/ml). Cell lysates were cleared from intact cells by centrifugation (5 min 1500 rpm, 4°C). Cleared cell lysates were TCA-precipitated and washed twice with acetone. Proteins were dissolved in water, lyophilized (in order to remove traces of organic

solvents) and solubilized in 100 mM $NH_4HCO_3$ (pH 8.0). Solubilization was attained by sonication for 3 × 40 s. Resolubilized proteins were reduced with dithiothreitol, alkylated with iodoacetamide, and in-solution digested with LysC (1:75 wt/wt) in 100 mM $NH_4HCO_3$ (pH 8.0). The resulting peptides were fractionated by reverse phase chromatography using an EC 250/4.6 Nucleosil 120-3 µm C18 column (Macherey–Nagel, Germany) and resulting fractions were analyzed by capillary electrophoresis-mass spectrometry (*Sarg et al., 2013*). Peptide separation was performed applying ultra low flow conditions (10 nl/min) using a neutral capillary installed into a PA800plus capillary electrophoresis system (Beckman Coulter, Germany), which was coupled via sheathless porous sprayer interface (*Faserl et al., 2011*) to an LTQ Orbitrap XL mass spectrometer (Thermo Scientific, Austria). Alternatively, cell lysates were fractionated in cytosolic and membrane associated proteins by ultracentrifugation (100.000×*g*) and proteins precipitated in 10% trichloroacetic acid (TCA). Precipitates were washed twice with acetone and resolubilized in SDS/urea sample buffer. SDS-PAGE in Tris-HCl gradient gels (4–15%, ReadyGel BioRad, Austria) was used to reduce sample complexity. The SDS gel was stained with SimplyBlue Safe Stain (Invitrogen, Austria), cut into 21 slices per lane and proteins in each slice were in-gel digested by Trypsin (50 ng/µl; biological sample 1) or LysC (50 ng/µl; biological sample 2). Resulting peptides were analyzed by liquid chromatography-mass spectrometry (LC -MS) using an UltiMate 3000 nano-HPLC system (Dionex/Thermo Scientific) coupled to an LTQ Orbitrap XL (Thermo Scientific). MS settings were as described (*de Godoy et al., 2008*).

## MS data analysis

MS data were analyzed using MaxQuant (Version 1.2.2.5) (*Cox and Mann, 2008*). The yeast ORF sequences from the Saccharomyces Genome Database (*Dwight et al., 2004*) were used for protein identification (last modified January 2010). The parameters for the enzymes, labels, maximum charge and variable modifications were chosen according to the experimental setup. All other settings were default. Quality control of the experiments was performed by comparing their label incorporation, peptide length distribution, calibrated and uncalibrated mass error distribution of retention time, fraction of matched MS/MS scans, and correlation of protein ratios between different replicates. To compare WT and *vps4Δ* mutants under rich conditions, we only used proteins with at least three heavy and light peptide counts. Each protein had at least one unique peptide and MS scans in at least two biological replicates. No additional filtering criteria were applied for the WT and *vps4Δ* data under starvation conditions. Differential regulation was estimated using the significance B (Perseus v1.0.2.13).

## GO enrichment analysis

The GO enrichment analysis was performed using the differentially regulated proteins. They were mapped against the GOSlim Generic biological processes and cellular components (*Harris et al., 2004*). GO term fusion was performed based on the GO tree (http://www.geneontology.org/). The enriched GO term at the highest level in the GO hierarchy was selected and its child terms were excluded. In cases when the parent process has a higher p-value, the child term was chosen. A hypergeometric test was used to estimate if the mapped GO term is significantly enriched with the selected proteins. The null hypothesis is that the selected proteins are randomly sampled from all yeast proteins. The resulting p-values were corrected with the Benjamini–Hochberg method. All adjusted p-values below 0.05 were reported. We also calculate the ratio of the observed (dataset frequency) vs the expected number of proteins (genome frequency) associated with the GO term, referred to as enrichment over genome (*McClellan et al., 2007*).

## Density plots and comparison of protein ratios across experiments

The density plots show the computed density estimates of the protein ratios quantified in all the three experiments, using Gaussian kernel (R software environment). The protein ratio distributions were compared pairwise using the Kolmogorov–Smirnov test, under the null hypothesis that the two tested groups are samples from the same distribution, that is, have the same median, variability and distribution shape. The significant protein changes were excluded. Additionally, paired Student's t test was used to test for mean difference in the log2-transformed protein ratios. All p-values were adjusted using the Benjamini–Hochberg test. To individually compare the ratios of each protein in WT and ESCRT mutant cells during starvation, we analyzed the ratio of ratios. For each protein quantified under the two starvation conditions, we took the ratio (fold change) of WT

and *vps4Δ* ratios and then transformed it on a log scale with base 2. To extract the most differentially regulated proteins, we calculated the z-scores from the normal distribution of the ratio of ratios and selected the critical values (i.e. those z-scores that are less likely to occur). The significance level was 0.05. The two-sided test resulted with 135 significant differences between WT and *vps4Δ* protein ratios.

## Statistical analysis

To explore and visualize the data we used the R language for statistical computing and graphics. To calculate the linear correlation between protein ratios we used Pearson correlation. All p-values below or equal to 0.05 were reported.

## Live cell fluorescence microscopy

A Zeiss Axio Imager M1 equipped with standard fluorescent filters and a SPOT Xplorer CCD camera was used. VisiView software was used for image-acquisition. Brightness and contrast were linearly adjusted. For vacuole staining (*Vida and Emr, 1995*) growing or starving cells were labeled for 10 min with 10 µg/ml FM4-64 (stock solution 1 mg/ml in DMSO), washed twice with and subsequently chased for 1 hr in the respective medium before microscopy was performed.

## Preparation of yeast whole cell protein extracts

To prepare whole cell lysates, yeast cells were pelleted, resuspended in ice-cold water with 10% trichloroacetic acid (TCA), incubated on ice for at least 30 min and washed twice with acetone. The precipitate was resolubilized in boiling buffer (50 mM Tris-HCl [pH 7, 5]; 1 mM EDTA, 1% SDS), solubilized with glass beads and boiled at 95°C. Urea sample buffer (150 mM Tris-HCl [pH 6, 8], 6 M Urea, 6% SDS, bromphenol blue, 10% β-mercaptoethanol) was added and the cleared cell lysate was separated by SDS-PAGE. Alternatively, proteins were extracted by alkaline extraction (*Kushnirov, 2000*). For analysis of protein phosphorylation (*Figure 3A,B*), TCA extraction was performed as described (*Papinski et al., 2014*).

## Western blot and immunodetection

Whole cell protein extracts were prepared by TCA extraction or alkaline lysis, separated by SDS-PAGE and transferred to PVDF membranes. Antibodies used in this study include: α-Flag (Sigma, Austria), α-GFP (IgG1K, Roche, Austria), α-Pgk (22C5D8, Life technologies, Austria), α-ALP (1D3A10, Life technologies), α-HA (12CA5, Abcam, UK), α-Pep12 (2C3G4, Abcam), α-CPY/Prc1 (clone 10A5, Invitrogen), α-Ape1 and α-Atg8 (*Papinski et al., 2014*). The α-Atg13 antibody was kindly provided by Daniel Klionsky, University of Michigan. α-Vps4 (*Babst et al., 1998*), α-Pep4 (*Klionsky et al., 1988*) and α-Vps21 (*Horazdovsky et al., 1994*) antibodies were kindly provided by Scott Emr, Cornell University. The α-Sch9 and α-Sch9pT737 antibodies were kindly provided by Robbie Loewith, University of Geneva.

## Alkaline phosphatase assays

The Pho8Δ60 assay was performed as described (*Noda et al., 1995*; *Klionsky, 2007*). Soluble endogenous vacuolar Pho8 activity (sPho8) was measured using a fluorigenic method described for Pho8Δ60 (*Noda and Klionsky, 2008*).

## Incorporation of [$^{35}$S]-Met/Cys

Mid-log cells grown in rich YNB medium (not containing methionine and cysteine) or starved cells were labeled with 15–30 µCi [$^{35}$S]-Met/Cys labeling mix (Hartmann analytics IS-103) for 5 min at 30°C and stopped with excess L-Met (5 mM) and 75 µg/ml cycloheximide. Cell extracts were analyzed by autoradiography of SDS-PAGE (Amersham Biosciences, Austria, STORM 840) or quantified by liquid scintillation counting (Beckmann Coulter LS6500).

## Amino acid extraction

Isogenic strains were grown to mid-log phase in YNB medium, washed twice with and inoculated in starvation medium at 0.6 OD$_{600nm}$/ml. Equal cell numbers (at least 30 OD$_{600nm}$) were harvested for each yeast strain and time point. Cultures from rich medium were harvested by vacuum filtration, washed twice with ice cold YNB medium (with 2% glucose and ammonium sulfate but without

amino acids) and twice with ice cold 60% methanol. Starving cultures were washed twice with ice cold 60% methanol. All cell pellets were air-dried over night and weighed. Amino acids were ethanol-extracted and analyzed as described (Altmann, 1992). Norvalin (8 nmol/mg dry weight) was the internal standard. For the representation of each individual amino acid, the measured values were normalized by the maximum measured amino acid content across all conditions and replicates.

### Bud index
Cells were briefly sonicated and visually scored for emerging buds by bright-field microscopy.

## Acknowledgements
We thank SD Emr, RN Collins and MD Erlacher for yeast strains and reagents and DJ Klionsky and R Loewith for antibodies. CK is supported by a 'Vienna Research Groups for Young Investigators' grant from the Vienna Science and Technology Fund (WWTF, VRG10-001) and a grant from the Austrian Science Foundation (FWF, P 25522-B20). The work was funded by HFSP-CDA-00001/2010-C, Austrian Science Fund (FWF), Y444-B12, MCBO (W01101), SFB021 (F21) to DT, and EMBO/Marie Curie (ALTF 642-2012; EMBOCOFUND2010, GA-2010-267146), MUI-START (2013042023) and a grant from the 'Tiroler Wissenschaftsfond' to OS.

## Additional information

### Funding

| Funder | Grant reference | Author |
| --- | --- | --- |
| Human Frontier Science Program (HFSP) | CDA-00001/2010-C | David Teis |
| Austrian Science Fund (FWF) | Y444-B12 | David Teis |
| Austrian Science Fund (FWF) | P 25522-B20 | Claudine Kraft |
| Austrian Science Fund (FWF) | MCBO (W01101) | David Teis |
| Austrian Science Fund (FWF) | SFB021 (F21) | David Teis |
| European Molecular Biology Organization (EMBO) and Marie Curie Actions cofunding | ALTF 642-2012 | Oliver Schmidt |
| European Molecular Biology Organization (EMBO) and Marie Curie Actions cofunding | EMBOCOFUND2010 | Oliver Schmidt |
| European Molecular Biology Organization (EMBO) and Marie Curie Actions cofunding | GA-2010-267146 | Oliver Schmidt |
| MUI START | 2013042023 | Oliver Schmidt |
| Tiroler Wissenschaftfonds | | Oliver Schmidt |
| Vienna Science and Technology Fund | WWTF, VRG10-001 | Claudine Kraft |

The funders had no role in study design, data collection and interpretation, or the decision to submit the work for publication.

### Author contributions
MM, Acquisition of data, Analysis and interpretation of data, Drafting or revising the article; OS, HL, DT, Conception and design, Acquisition of data, Analysis and interpretation of data, Drafting or revising the article; MA, ZT, Analysis and interpretation of data, Drafting or revising the article; KF, SW, LK, TP, TD, Acquisition of data, Analysis and interpretation of data; CK, Conception and design, Analysis and interpretation of data, Drafting or revising the article

### Author ORCIDs
David Teis, http://orcid.org/0000-0002-8181-0253

# Additional files

## Supplementary files

- Supplementary file 1. Quantification of proteome changes in WT cells during amino acid starvation.

- Supplementary file 2. GO Analysis of significantly up- or down-regulated proteins under starvation.

- Supplementary file 3. Quantification of proteome changes in WT cells and *vps4Δ* mutants during starvation.

- Supplementary file 4. Quantitative proteomics comparing WT and *vps4Δ* under rich growth conditions.

- Supplementary file 5. 135 most differentially regulated proteins in WT and *vps4Δ* mutants during starvation.

- Supplementary file 6. GO Analysis of proteins that are differentially regulated under starvation.

- Supplementary file 7. Yeast strains and plasmids.

- Supplementary file 8. Primer sequences.

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
