## [Decision Letter]

Thank you for sending your work entitled “The sequential action of the MVB pathway and autophagy ensures cell survival during starvation” for consideration at *eLife*. Your article has been favorably evaluated by Vivek Malhotra (Senior editor), Noboru Mizushima (Reviewing editor), and two reviewers.

The Reviewing editor and the reviewers discussed their comments before we reached this decision, and the Reviewing editor has assembled the following comments to help you prepare a revised submission.

The manuscript of Mueller et al. focuses on the relative contribution of the MVB pathway to cellular amino acid homeostasis during starvation. The authors reveal that cells act in a consecutive manner to mobilize amino acids from cytosolic proteins (presumably via the proteasome), membrane proteins (via the MVB pathway), and finally bulk cytosolic proteins via autophagy. Taking a proteomic approach to compare MVB mutant and wild-type cells, the authors show that starvation is coupled to a massive remodeling of the proteome, where multiple hydrolases are upregulated to allow degradation of membrane proteins. The authors conclude that the MVB pathway is critical to supply amino acids for survival of cells during the early phase of starvation, which is followed by autophagy activation.

Some of these observations on MVB were previously made (Jones et al. Traffic 2012), whereas Müller et al. take a more global comparative approach by following both MVB and autophagy pathway in parallel. Overall, we think that this paper is important and seems mostly sounds.

Major comments:

1) The data showing the sequential activation of the MVB pathway and autophagy during starvation is not strong. Autophagy can be immediately initiated after starvation. It is recommended to use GFP-Atg8 as another autophagy marker in addition to Rpl25-GFP in Figure 1, which may show faster kinetics. Indeed, free GFP is clearly detected at 2h after starvation in Figure 6.

2) It is also essential to show that the amino acid pool is indeed reduced in *vps4∆* cells during early starvation. It was previously shown that the amino acid levels are reduced in autophagy mutants compared to wild-type cells at as early as 3h after starvation (Onodera and Ohsumi, JBC 2005). A similar experiment should be performed using MVB, autophagy, and double mutant cells, which will provide important information on relative contributions of these two pathways, in terms of both time and quantity, to the amino acid pool during starvation.

3) Although the phenotype of *vps4∆* cells suggests that starvation-induced cargo degradation in the MVBs is important to supply amino acids for synthesis of vacuolar enzymes, it is still possible that, in *vps4∆* cells, delivery of hydrolases to the vacuole is defective, which affects the vacuolar function independently of amino acid supply. These two possibilities are not clearly distinguished. In particular, the data shown in Figures 4 and 6 suggest that there may be a defect in the vacuolar function even in growing conditions or during a very short starvation period in *vps4∆* cells. This needs to be further clarified to support the authors' main conclusion.

4) The data in Figure 5 should be quantified because, as the authors mention in Discussion, several papers have reported that the ESCRT machinery is important for a late step of autophagy (i.e. autolysosome formation).

5) The authors mention that selected membrane proteins are degraded during starvation. The methionine permease Mup1 is certainly degraded very efficiently (Figure 1). Do they also have examples of membrane proteins that are unaffected? Can they discuss how the cell can distinguish which proteins are degraded selectively?

6) In the same experiment, do the authors use Met/Cys free medium when they label cells under nutrient rich conditions? Otherwise, relative activity of 35S-Met/Cys would be different between the rich and starved conditions.

7) The ESCRT-mediated MVB formation would be important for degradation of not only membrane proteins but also cytosolic proteins that are engulfed inside intraluminal vesicles. This is a reminiscence of microautophagy. Indeed, it was reported by Cuervo's lab that MBV formation contributes to the degradation of cytosolic proteins (Sahu et al. Dev Cell. 2011, 20:131). This possibility should also be discussed.

[Editors’ note: the decision after re-review follows, after which the authors submitted for further consideration.]

Thank you for sending your revised work entitled “The sequential action of the MVB pathway and autophagy ensures cell survival during starvation” for consideration at *eLife*. Your full submission has been evaluated by Vivek Malhotra (Senior editor), a Reviewing editor, and one of the original reviewers.

The authors have responded to reviewers concerns by including new data, but major concerns still remain that preclude the publication of the paper at this time. The concerns of the reviewers follow.

1) The data presented on GFP-Atg8n in new Figure 1 together with Figure 1—figure supplement 1 shows a relative rapid induction in autophagy activity during starvation. Thus, it is inappropriate to suggest: “the degradation of cytoplasmic contents via autophagy passed through an initial lag phase and peaked later during starvation, when compared to membrane protein degradation via the MVB pathway (Figure 1)”. The time at which degradation of a protein peaks depends not only on the degradation activity, but also its rate of synthesis and total expression level. The authors have used selective cargos to monitor MVB pathway, it is also important to use selective cargos (e.g. GFP-Atg8) to monitor autophagy and discuss their temporal relationship and relative contributions. The response of the authors to this concern is not acceptable, for example, the expression levels of Atgs are not direct indicators of autophagic activity.

2) The authors show increased amino acid levels in wild-type cells during starvation. This is puzzling and inconsistent with a previous report (Onodera and Ohsumi, JBC 2005), where the amino acid concentration was dramatically reduced even in wild-type cells and more profoundly in autophagy mutant cells. The authors do not discuss this apparent discrepancy. This should be addressed experimentally.

3) The data in Figure 4 reveals a clear defect in processing of mPho8 into sPho8 even under growing conditions. This indicates a defect in vacuolar hydrolytic activity. Indeed, the authors describe a partial sorting defects of vacuolar hydrolases in ESCRT mutants and they have tried to rescue them by over expression of Vps10 (Figure 4). However, this was performed only for data shown in Figure 4 and Figure 7—figure supplement 1. Vps10 rescue experiments should be included in other experiments that measure the amino acid pool (Figure 2), protein synthesis (Figure 2), and autophagic activity (Figure 6). These are more relevant to the potential function of the MVB pathway as an amino acid generator. It is also unclear why over expression of Vps10 cannot rescue the pho8 activity under starvation conditions. Taken together, it is likely that defects in protein turnover observed in ESCRT mutants are due to a combination of defects in sorting of vacuolar enzymes and generation of amino acids.

---

## [Author Response]

*1) The data showing the sequential activation of the MVB pathway and autophagy during starvation is not strong. Autophagy can be immediately initiated after starvation. It is recommended to use GFP-Atg8 as another autophagy marker in addition to Rpl25-GFP in*
Figure 1*, which may show faster kinetics. Indeed, free GFP is clearly detected at 2h after starvation in*
Figure 6.

We never intended to argue that autophagy is only activated after membrane proteins were degraded via the MVB pathway. Therefore we have also slightly modified our model in Figure 7 (upper panel) to avoid possible misunderstanding. Starvation induced cargo ubiquitination and autophagy are probably simultaneously induced once TORC1 signaling is switched off due to nutrient limitation, as discussed in the original manuscript. Yet, while the constitutive MVB pathway can readily degrade ubiquitinated membrane proteins, autophagy cannot immediately operate at maximal capacity. Throughout starvation, there is a gradual increase in the magnitude, strength and efficiency of non-selective autophagy. It takes time and cellular resources to ramp up the autophagic machinery until it efficiently captures bulk cytoplasm for efficient amino acid recycling. All our data support this concept:

A) The delivery of cytoplasm by the autophagic machinery gradually increases during extended starvation. This is a well-accepted concept and supported using a Pho8 60 assay, a sensitive method to measure autophagy early during starvation (Figure 1—figure supplement 1) (Noda, T. et al. 1995). Pho8 60 activity increases in the first 3 hours of starvation, indicating the onset of autophagy. Yet the major increase is observed later after overnight starvation (Figure 1—figure supplement 1). This is also consistent with the autophagic delivery of highly abundant cytoplasmic proteins such as ribosomal subunits (Figure 1) or Fba1 (1.000.000 molecules/cell) (Figure 1—figure supplement 1).

B) Our new results demonstrate that at the onset of starvation, GFP-Atg8 is approximately 10–20 times less abundant and almost not detectable when directly compared to Mup1-GFP (Figure 1). Both constructs are expressed from their endogenous promoters. Some Atg8 is transported into the vacuole about 2-3 hours after starvation (Figure 1) and overall Atg8 protein levels strongly increase to enhance autophagy during starvation (Figure 1, Figure 1—figure supplement 1), Kirisako, T. 1999 JCB). In contrast Mup1-GFP and other membrane proteins (e.g.: Can1, Ste2, Fur4) are almost completely degraded after 3 hours of starvation (Figures 1 and 3).

C) These findings are also consistent with the idea that higher Atg8 protein levels allow to grow larger phagophores that would engulf increasing amounts of bulk cytoplasm (Xie Z. et al. 2008, MBoC; Abelovich H. et al. 2000, JCB).

D) This induction of the autophagic machinery is also reflected by our quantitative proteomic data. The protein levels of Atg8 (3.7 fold increase) and other autophagy core components such as Atg1 (3.3 fold increase), Atg3 (1.5 fold increase), Atg7 (1.4 fold increase) and Atg15 (2.5 fold induction) increase during starvation (Figure 3, Figure 3–source data 1).

E) The protein levels of the selective autophagic cargo Ape1 strongly increase early during starvation (Figures 4 and 6). Similarly, the protein levels of Ams1, another selective autophagic cargo molecule, increase during starvation. Hence early clipping of Atg8, at least in part, also indicates an increase of the Atg19> dependent biosynthetic delivery of pApe1 and pAms1 to the vacuole alongside bulk autophagy (Scott et al. 2001 Mol. Cell; Sawa-Makarska J,. Nat. Cell. Biol. 2014).

F) All of this is consistent with our results presented in Figure 1. Even highly abundant autophagy substrates such as two different ribosomal subunits (Rps2 and Rpl25) are not immediately captured by autophagy and only efficiently delivered into vacuoles upon extended periods of starvation. Similar results were obtained using Fba1-GFP, which is one of the most abundant cytoplasmic proteins (app. 1.000.000 molecules / cell; Ghaemmaghami S, Nature 2003) (Figure 1—figure supplement 1).

G) Finally our analysis of amino acids levels during starvation demonstrates that the MVB pathway provides amino acids prior to autophagy (Figure 2, see below).

Taken together, all our data support the idea that the MVB pathway and autophagy contribute sequentially to cell survival during nutrient limitation. Selective protein degradation pathways (UPS and MVB pathway) bridge the gap in amino acid recycling before an efficient autophagic response develops.

*2) It is also essential to show that the amino acid pool is indeed reduced in* vps4*∆ cells during early starvation. It was previously shown that the amino acid levels are reduced in autophagy mutants compared to wild-type cells at as early as 3h after starvation (Onodera and Ohsumi, JBC 2005). A similar experiment should be performed using MVB, autophagy, and double mutant cells, which will provide important information on relative contributions of these two pathways, in terms of both time and quantity, to the amino acid pool during starvation*.

We have measured free amino acid levels two and four hours after starvation in WT cells, ESCRT mutants (*vps4*), autophagy mutants (*atg8*) and the double mutant (*vps4*, *atg8*) (Figure 2). Under rich growth conditions the amino acid levels of WT cells, ESCRT mutants, autophagy mutants and the double mutants are comparable. Already two hours after starvation the levels of free amino acids were lower in ESCRT mutants when compared to WT cells or autophagy mutants. The double mutant always showed the strongest reduction in amino acid levels (Figure 2).

*3) Although the phenotype of* vps4*∆ cells suggests that starvation-induced cargo degradation in the MVBs is important to supply amino acids for synthesis of vacuolar enzymes, it is still possible that, in* vps4*∆ cells, delivery of hydrolases to the vacuole is defective, which affects the vacuolar function independently of amino acid supply. These two possibilities are not clearly distinguished. In particular, the data shown in*
Figures 4 and 6
*suggest that there may be a defect in the vacuolar function even in growing conditions or during a very short starvation period in* vps4*∆ cells. This needs to be further clarified to support the authors' main conclusion*.

The graph in Figure 4 displays the starvation-induced up-regulation of vacuolar hydrolases in WT and ESCRT mutants. The results show that ESCRT mutants cannot efficiently upregulate vacuolar hydrolases during starvation. Yet, under rich conditions, the total protein levels of the vacuolar master peptidase Pep4 were only marginally decreased in ESCRT mutant cells (Figure 4—figure supplement 1). Consistently under rich and starvation conditions, pPho8, which reaches the vacuole via the AP3 pathway, efficiently matured into mPho8 in WT and ESCRT mutants in a Pep4 dependent manner. pPho8 accumulated in *pep4∆, prb1∆, prc1∆* mutants but not in ESCRT mutants; Figure 4, lanes 1-5. Yet, the subsequent starvation-induced processing of mPho8 into sPho8 is strongly hampered in ESCRT mutants (Figure 4 and Figure 4—figure supplement 1).

While there is certainly a subtle maturation defect of Ape1 in ESCRT mutants, not only pApe1 but also mApe1 protein levels are increased under rich conditions, so that overall Ape1 activity is compensated for (Figure 6). This is in part also due to higher mRNA levels of Ape1 (and also Prb1) under rich conditions, which we determined in separate gene-expression profiling experiment to compare WT cells and ESCRT mutants. Similarly, Atg8 protein (Figure 6) and mRNA levels are increased in ESCRT mutants. None of these results suggest a strong defect in vacuolar function under rich conditions. The results are more consistent with a partial and relatively minor sorting defect of vacuolar hydrolases.

In addition, we now specifically rescued the partial sorting defects of vacuolar hydrolases in ESCRT mutants by over-expression of Vps10 (Figure 4—figure supplement 1). Vps10 is the sorting receptor for multiple vacuolar hydrolases. Consistent with previous reports (Babst et al., Embo J., 1998), Vps10 over-expression rescued the partial mis-sorting of vacuolar hydrolases (CPY/Prc1, Pep4) in ESCRT mutants (*vps4*) and hence restored the catabolic activity of vacuoles to >80% of WT levels (Figure 4 and Figure 4—figure supplement 1) but not the degradation of membrane proteins. Even though the catabolic activity of ESCRT mutants was restored under rich conditions, ESCRT mutants still failed to boost the catabolic activity of their vacuoles under starvation, and subsequently stopped to grow and failed to enter quiescence (Figure 7—figure supplement 1). Moreover, loss of Vps10 caused a strong sorting defect for vacuolar hydrolases (but not a complete block), without affecting membrane protein delivery into vacuoles via the MVB pathway (Figure 4—figure supplement 1) (Whyte, JR. Curr. Biol. 2001). Yet, *vps10* single mutants had more modest effects on cell growth during starvation than loss of the ESCRT pathway, which could be complemented by overexpression of Vps10 (Figure 7—figure supplement 1). These findings reject a mere role of the MVB pathway for vacuolar sorting of hydrolases, and instead support the idea that selective membrane protein degradation significantly contributes to maintaining cell homeostasis during starvation.

*4) The data in*
Figure 5
*should be quantified because, as the authors mention in Discussion, several papers have reported that the ESCRT machinery is important for a late step of autophagy (i.e. autolysosome formation)*.

Quantification of the localization of pHluorin-Atg8 in >120 cells demonstrates that pHluorin-Atg8 fluorescence was quenched in the vacuoles of most WT cells and ESCRT (>90%) mutants (Figure 5). In contrast, pHluorin-Atg8 remained fluorescent in the vacuoles of >90% of *pep4∆, prb1∆, prc1∆* and *vma4∆* mutant cells.

*5) The authors mention that selected membrane proteins are degraded during starvation. The methionine permease Mup1 is certainly degraded very efficiently (*Figure 1*). Do they also have examples of membrane proteins that are unaffected? Can they discuss how the cell can distinguish which proteins are degraded selectively*?

The general amino acid permease Gap1 is strongly induced during starvation and transported to the cell surface while other amino acid permeases such as Mup1 or Can1 are endocytosed and degraded (Figure 3, Figure 3—figure supplement 2B, C). Interestingly the up -regulation of Gap1 appears to be temporally coordinated with the down-regulation of other membrane proteins and is stronger in WT as compared to ESCRT mutants, consistent with overall reduced proteome remodeling. In addition to Gap1, our quantitative proteomics data also suggest that Mep2 is upregulated and that the total levels of Pma1 do not change significantly (Figure 3, Figure 3–figure supplement 2A, Figure 3–source data 2). After some time a portion of Gap1-GFP and Pma1-GFP is also transported to the vacuole by steady state turnover in quiescent cells, but the majority of these membrane proteins remains at the cell surface.

How growing cells or starving cells select membrane proteins for degradation is currently not entirely clear but probably involves signal transduction-mediated regulation (TORC1/Npr1) of specific adaptors for the HECT-type ubiquitin ligase RSP5. We have included this possibility in the Discussion. In addition we have new data that suggest that distinct ubiquitination sites on Mup1 are required for starvation- vs. ligand-induced endocytosis, but this would go beyond the scope of this study.

*6) In the same experiment, do the authors use Met/Cys free medium when they label cells under nutrient rich conditions? Otherwise, relative activity of 35S-Met/Cys would be different between the rich and starved conditions*.

We have used Met/Cys free medium for all our experiments and described the media in the Material and methods section.

*7) The ESCRT-mediated MVB formation would be important for degradation of not only membrane proteins but also cytosolic proteins that are engulfed inside intraluminal vesicles. This is a reminiscence of microautophagy. Indeed, it was reported by Cuervo's lab that MBV formation contributes to the degradation of cytosolic proteins (Sahu et al. Dev Cell. 2011, 20:131). This possibility should also be discussed*.

Thank you for pointing this out. We have changed the text in the Discussion accordingly: **“**Interestingly, the ESCRT machinery has recently been described to contribute to selective micro-autophagy on late endosomes in mammalian cells (58). Hence, ESCRT-dependent catabolic pathways may not be limited to membrane proteins, although our study did not experimentally address this possibility.”

[Editors’ note: the manuscript was accepted after being submitted for further consideration by the original reviewers.]

In the past 12 weeks we have experimentally addressed and clarified all concerns of the reviewers. The detailed answers are attached on the following pages.

Summary of key experiments:

a) We thoroughly analyzed the onset of autophagy using GFP-Atg8 in relation to other selective and non-selective autophagic cargoes as well as MVB cargo and we discuss their temporal relationships. Our results are consistent with the essential role of autophagy for amino acid recycling later during starvation (see point b).

b) We have improved the amino acid analysis. In WT cells, amino acid levels decrease to about 50% during the first 2 hours of starvation and fully recover in an autophagy-dependent manner at around 4 hours of starvation. Most importantly, our analysis also shows that the MVB pathway is essential to maintain free intracellular amino acid pools in the first 2 hours of starvation.

c) We have measured the amino acid pool, protein synthesis and autophagic activity in *vps4∆* mutants that overexpress Vps10. The results are consistent with the idea that selective membrane protein degradation via the MVB pathway, rather than the mere sorting of vacuolar hydrolases, is essential to maintain sufficient free intracellular amino acids to uphold protein synthesis for proteome remodeling early during starvation.

To our mind these results further strengthen our model: early during starvation, the degradation of membrane proteins via the MVB pathway is required to maintain a critical intracellular amino acid pool for the synthesis of new proteins, including vacuolar hydrolases to boost the catabolic activity of vacuoles. This primes vacuoles for the efficient degradation of bulk cellular material via autophagy, which is essential to enter a stable G1/G0-arrested, quiescent state. Thus, a coordinated action of the MVB pathway and autophagy is key to survive extended periods of nutrient limitation.

*1) The data presented on GFP-Atg8n in new*
Figure 1
*together with*
Figure 1—figure supplement 1
*shows a relative rapid induction in autophagy activity during starvation. Thus, it is inappropriate to suggest* “*the degradation of cytoplasmic contents via autophagy passed through an initial lag phase and peaked later during starvation, when compared to membrane protein degradation via the MVB pathway (*Figure 1*)*”*. The time at which degradation of a protein peaks depends not only on the degradation activity, but also its rate of synthesis and total expression level. The authors have used selective cargos to monitor MVB pathway, it is also important to use selective cargos (e.g. GFP-Atg8) to monitor autophagy and discuss their temporal relationship and relative contributions. The response of the authors to this concern is not acceptable, for example, the expression levels of Atgs are not direct indicators of autophagic activity*.

We thank the reviewers for critically pointing out that our statement ‘autophagy passed through a lag phase ’ was mis-leading, especially since we never intended to argue that the MVB pathway and autophagy are sequentially activated. Apparently, our intended message, namely that the MVB pathway has an earlier impact on the cellular amino acid levels when compared to autophagy, was not clearly put into words. To avoid this, we have now replaced ‘sequential action*’* with ‘coordinated action’ throughout the manuscript and also in the title.

In addition, we have performed the recommended experiments and used GFP-Atg8 as an additional marker to determine the vacuolar delivery of all selective and non-selective autophagosomes as well as cvt-vesicles (Figure 1). Small amounts of free GFP released from GFP-Atg8 inside vacuoles could be readily detected by western blot analysis 1 hour after the onset of starvation and the levels of free GFP strongly increased at 3 hours of starvation (Figure 1). These findings are consistent with the de novo synthesis and the overall increase of endogenous Atg8 during starvation (Figure 1—figure supplement 1) as observed earlier by the Ohsumi lab (27). Work from the Klionsky lab (1; 77) further demonstrated that this increase in Atg8 protein levels was required for the formation of larger (but not more) autophagosomes that could capture larger volumes of cytoplasm during ongoing starvation. Our results showing the early degradation of GFP-Atg8 and the continuous increase in autophagic degradation of highly abundant selective (Rps2, Rpl25) (Figure 1) and non-selective (Fba1, Pho8 60) (Figure 1—figure supplement 1) cargoes throughout starvation are fully compatible with this model. Furthermore, our findings are consistent with the essential role of autophagy for amino acid recycling later during starvation (see point b and Figure 2).

In summary our results suggest that both autophagy and starvation induced-endocytosis are simultaneously activated early during starvation (probably by inactivation of TORC1). Autophagy continuously delivers ever-increasing amounts of cytoplasm into vacuoles and remains active during extended periods of starvation, whereas starvation-induced degradation of membrane proteins via the MVB pathway appears to be completed relatively early (within the first 4 hours of starvation). Our analysis of intracellular amino acid recycling supports this idea and demonstrates that autophagy is essential for amino acid recycling later during starvation, whereas the MVB pathway appears to be more important to maintain intracellular amino acid levels early during starvation (see reply to concern 2 and Figure 2).

*2) The authors show increased amino acid levels in wild-type cells during starvation. This is puzzling and inconsistent with a previous report (Onodera and Ohsumi, JBC 2005), where the amino acid concentration was dramatically reduced even in wild-type cells and more profoundly in autophagy mutant cells. The authors do not discuss this apparent discrepancy. This should be addressed experimentally*.

This point was well taken and we now managed to rapidly harvest yeast cells in less than one 1 minute by filtration and thereby avoided starvation conditions during sample preparation. We measured the intracellular levels of 18 different amino acids in WT cells, MVB (*vps4)* or autophagy (*atg8*) mutants or double mutants (*vps4, atg8*) by HPLC. 1 hour after starvation, the total free amino acid pool decreased to similar levels in WT cells and all mutant strains (Figure 2). In WT cells the levels of most amino acids continued to decrease for another hour to about 50% of the levels measured under rich conditions. At around 4 hours of starvation the overall levels of amino acids almost fully recovered in an autophagy-dependent manner. These results are at large consistent with previous findings (49), where a decrease to approximately 30% of intracellular amino acid levels was detected during the first two hours of starvation and autophagy was required for the partial recovery of amino acid levels from 3 to 6 hours of starvation. We also discuss these differences in the text.

Importantly, our amino acid analysis from *vps4* mutants showed that the MVB pathway essentially contributed to maintain the overall levels of free intracellular amino acids at 2 hours of starvation. At this time point, the levels of 14 individual amino acids were lower in *vps4* mutants when compared to WT cells or autophagy mutants and failed to recover during extended starvation (Figure 2).

These results showed that the MVB pathway was essential to maintain the levels of most free intracellular amino acids in the first 2 hours of starvation, while autophagy was essential to restore intracellular amino acids later during starvation.

*3) The data in*
Figure 4
*reveals a clear defect in processing of mPho8 into sPho8 even under growing conditions. This indicates a defect in vacuolar hydrolytic activity. Indeed, the authors describe a partial sorting defects of vacuolar hydrolases in ESCRT mutants and they have tried to rescue them by over expression of Vps10 (*Figure 4*). However, this was performed only for data shown in*
Figure 4
*and*
Figure 7—figure supplement 1*. Vps10 rescue experiments should be included in other experiments that measure the amino acid pool (*Figure 2*), protein synthesis (*Figure 2*), and autophagic activity (*Figure 6*). These are more relevant to the potential function of the MVB pathway as an amino acid generator. It is also unclear why over expression of Vps10 cannot rescue the pho8 activity under starvation conditions. Taken together, it is likely that defects in protein turnover observed in ESCRT mutants are due to a combination of defects in sorting of vacuolar enzymes and generation of amino acids*.

Vps10 over-expression in *vps4* mutants rescued the partial sorting defects for vacuolar hydrolases but not membrane protein degradation. Consistently, Vps10 over-expression almost completely restored the catabolic activity of vacuoles under rich conditions (Figure 5, Figure 5—figure supplement 1). Our new experiments further demonstrate that *vps4* mutants overexpressing Vps10 still failed to maintain intracellular amino acid levels and protein synthesis during starvation (Figure 5—figure supplement 1; Figure 5—figure supplement 2A, B). Therefore *vps4* mutants over-expressing Vps10, similar to *vps4* mutants alone, could not efficiently up-regulate the de novo synthesis of vacuolar hydrolases and thereby boost the catabolic activity of vacuoles (Figure 5), which in turn prevented the efficient proteolytic conversion from mPho8 into sPho8 (Figure 5). In consequence autophagic cargo (Rpl25-GFP) was not efficiently degraded (Figure 6—figure supplement 1). In addition, we determined the subcellular localization of the vacuolar master protease Pep4-GFP and Prc1/CPY-RFP in ESCRT mutants under rich growth or starvation using live cell fluorescence microscopy. Pep4-GFP localized to the class E compartment in ESCRT mutants, but a large fraction of Pep4-GFP was also delivered into the lumen of the vacuole (Figure 5). Similar results were obtained for Prc1/CPY-RFP (Figure 5—figure supplement 1). In the lumen of the vacuole the hydrolases matured and were catabolically active (Figures 3 and 5). Most likely therefore ESCRT mutants were never isolated as *pep* mutants, unlike other endo-lysosomal trafficking complexes such as HOPS, CORVET or retromer ([24], Genetics).

Hence the striking cellular defects of ESCRT mutants during starvation can be mostly attributed to their block in membrane protein degradation, rather than the mere and rather partial sorting defects for vacuolar hydrolases.